



# The Last Glacial Maximum in central North Island, New Zealand: palaeoclimate inferences from glacier modelling

Shaun R. Eaves[1,2], Andrew N. Mackintosh[1,2], Brian M. Anderson[1], Alice M. Doughty[3], Dougal B. Townsend[4], Chris E. Conway[2], Gisela Winckler[5], Joerg M. Schaefer[5], Graham S. Leonard[4], and Andrew T. Calvert[6]

[1]Antarctic Research Centre, Victoria University of Wellington, PO Box 600, 6140 Wellington, New Zealand
[2]School of Geography, Earth and Environmental Science, Victoria University of Wellington, PO Box 600, 6140 Wellington, New Zealand
[3]Department of Earth Science, Dartmouth College, Hanover, NH 03755, USA
[4]GNS Science, 1 Fairway Drive, Avalon, PO Box 30-368, Lower Hutt 5040, New Zealand
[5]Lamont–Doherty Earth Observatory, Columbia University of New York, Palisades, NY 10964, USA
[6]Volcano Science Center, US Geological Survey, Menlo Park, CA 94025, USA

*Correspondence to:* Shaun R. Eaves (shaun.eaves@vuw.ac.nz)

**Abstract.** Quantitative palaeoclimate reconstructions provide data for evaluating the mechanisms of past, natural climate variability. Geometries of former mountain glaciers constrained by moraine mapping afford the opportunity to reconstruct palaeoclimate, due to the close relationship between ice extent and local climate. In this study, we present results from a series of experiments using a 2D coupled energy-balance/ice-flow model that investigate the palaeoclimate significance of Last Glacial Maximum moraines within nine catchments in central North Island, New Zealand. We find that the former ice limits can be simulated when present day temperatures are reduced by between 4 °C and 7 °C, when precipitation remains unchanged from present. The spread in the results between the nine catchments is likely to represent the combination of chronological and model uncertainties. The temperature decrease required to simulate the former glaciers falls in the range of 5.1 °C and 6.3 °C for the majority of catchments targeted, which represents our best estimate of the peak temperature anomaly in central North Island, New Zealand during the Last Glacial Maximum. A decrease in precipitation, as suggested by proxy evidence and climate models, of up to 25 % from present, increases the magnitude of the required temperature changes by up to 0.8 °C. Glacier model experiments using reconstructed topographies that exclude the volume of post-glacial (<15 ka) volcanism, generally increased the magnitude of cooling required to simulate the former ice limits by up to 0.5 °C. Our palaeotemperature estimates expand the spatial coverage of proxy-based quantitative palaeoclimate reconstructions in New Zealand, and are consistent with independent, proximal temperature reconstructions from fossil pollen assemblages, as well as similar glacier modelling reconstructions from central Southern Alps.



# 1 Introduction

The Last Glacial Maximum ('LGM') describes the global sea level low-stand at 26 - 19 ka, when Northern Hemisphere ice sheets reached their maximum extent of the last glacial cycle (Clark et al., 2009). During this interval, global mountain glacier extent also peaked (Schaefer et al., 2006; Clark et al., 2009). In New Zealand, local nomenclature such as 'extended LGM' (Newnham et al., 2007) or more recently, 'Last Glacial Cold Period' (Alloway et al., 2007; Barrell et al., 2013) has been introduced to describe the glacial climatic conditions that prevailed c. 30 - 18 ka (e.g. Vandergoes et al., 2005; Putnam et al., 2013b). Continuous and well-dated climate proxy records have greatly improved understanding of the timing and relative magnitudes of climatic changes in New Zealand through this period (e.g. Barrell et al., 2013). However, quantitative estimates of terrestrial palaeoclimatic variables (namely air temperature and precipitation) are rare. Where available, independent estimates of LGM climate have shown good agreement across relatively short spatial scales (e.g. central Southern Alps - Golledge et al., 2012; Putnam et al., 2013a, b; Rowan et al., 2013). However, quantitative palaeoclimate reconstructions from elsewhere in New Zealand can differ greatly. For example, McKinnon et al. (2012; their Table 3) summarise all previously published, terrestrial, LGM temperature estimates for New Zealand, which range from +0.5°C to -9°C, relative to present. Such differences may arise from methodological and/or chronological uncertainties, or could represent meaningful spatial relationships that represent key climatic processes (e.g. Lorrey et al., 2012) related to the drivers of change. Increasing the number and spatial coverage of quantitative palaeoclimate reconstructions will help to resolve these issues.

In central North Island New Zealand, geological evidence, primarily in the form of moraines, suggests that small ice-fields or ice caps existed on Ruapehu and Tongariro volcanoes during the late Quaternary (Mathews, 1967; McArthur and Shepherd, 1990; Barrell, 2011). Recent cosmogenic [3]He surface exposure dating of moraine boulders (Eaves et al., 2016; Eaves, 2015) on both volcanoes, supported by a local production rate calibration (Eaves et al., 2015), confirms that these ice masses existed during the LGM. Geometric reconstructions of these ice masses suggest that local equilibrium line altitudes (ELAs) were depressed by c. 1000 m relative to present (McArthur and Shepherd, 1990; Eaves et al., 2016). However, these reconstructions are hampered by the paucity of geomorphic evidence to constrain past ice thickness on the upper mountain, which can lead to errors in manual ELA reconstruction (Rea et al., 1999). Furthermore, localised topographic changes resulting from effusive post-glacial volcanism (Hobden et al., 1996), as well as post-glacial flank collapse (Palmer and Neall, 1989; Eaves et al., 2015), also contribute uncertainty to glacier reconstructions. In this paper, we use numerical glacier model experiments to investigate the LGM climate of central North Island, New Zealand, in order to answer the following questions:

– What are the temperature and precipitation changes, relative to present, required to simulate the LGM ice masses in central North Island?

– How do post-glacial topographic changes on the volcanoes influence our glacier model based estimates of LGM climate?



## 2 Setting

### 2.1 Geology and climate of the central North Island volcanoes

Located in central North Island, New Zealand (39.2 °S 175.6 °E; Fig. 1), Tongariro Volcanic Centre (TgVC) represents the southernmost expression of the Taupo Volcanic Zone, which is a c. 300 km long, northeast trending belt of subduction zone volcanism at the Australian-Pacific plate margin (Cole, 1978). TgVC is dominated by the andesite-dacite stratovolcanic centres of Tongariro massif (including Mt. Tongariro - 1963 m above sea level (asl) and the Holocene cone of Mt. Ngauruhoe - 2291 m asl) and Mt. Ruapehu (2797 m asl). Radiometric dating of lava flows indicates that cone-building volcanism in the region began before c. 275 ka (Stipp, 1968; Gamble et al., 2003), and both centres have exhibited effusive volcanic activity in historical times (Houghton et al., 1987).

At 2797 m asl, Mt. Ruapehu is the highest peak in North Island and the only to intercept the present day end of summer snowline. Several small ($< 1$ km$^2$) cirque glaciers currently exist on the upper mountain slopes (Fig. 1). The average annual ELA for the five main cirque glaciers on Mt. Ruapehu is c. 2500 m (Keys, 1988). Due to its lower elevation, no glacial ice is present on Tongariro massif today.

The local climate, as recorded at Whakapapa Village (1100 m asl; Fig. 1B) on the northwest flank of Mt. Ruapehu, is characterised by high annual precipitation (c. 2700 mm a$^{-1}$), with low seasonal variability. For example, winter (JJA) precipitation averages c. 760 mm, compared to c. 620 mm in summer. Monthly mean temperatures at this altitude range from c. 13 °C in February to c. 3 °C in July, with an annual average of 7.5 °C (2000-2010; NIWA, 2014).

### 2.2 LGM ice limits in central North Island

Abundant geomorphic evidence for glacial erosion and deposition, which exists as much as several kilometres down valley of the present day ice limits, has long been recognised in central North Island (Taylor, 1926; Mathews, 1967; McArthur and Shepherd, 1990; Barrell, 2011). For Mt. Ruapehu, two independent mountain-scale assessments of moraine distribution have produced mapped reconstructions of a former late Pleistocene ice mass (McArthur and Shepherd, 1990; Barrell, 2011), which are in good agreement with one another (Fig. 1A). These reconstructions are constrained by moraines and suggest that an ice mass centred over the volcano summit was drained by four main valley glaciers that terminated between 1100 m and 1200 m asl in the Mangatoetoenui (MTO), Wahianoa (WAH), Mangaturuturu (MTU) and Whakapapanui (WHA) valleys (Fig. 1A; Table 1). The moraine evidence indicates that smaller lobes of ice existed between these main outlets, however the former ice geometries in these locations are less certain, which is reflected by the higher frequency of disagreement between the two reconstructions (Fig. 1A).

Only one mountain-scale palaeoglacial reconstruction of a late Pleistocene ice mass has been undertaken for Tongariro massif





(Barrell, 2011). This work depicts multiple valley glaciers draining the western and eastern flanks of the volcano, which terminated between 1100 m and 1400 m asl (Fig. 1A). This reconstruction is in good agreement with the moraine mapping and geomorphic descriptions given by Mathews (1967) and Eaves et al. (2016).

Until recently, age constraint for the moraines in TgVC has come from tephrostratigraphy and correlation to better-dated glacial sequences in the Southern Alps. For example, Topping (1974) tentatively identified the Kawakawa/Oruanui Tephra (KOT; 25.4 ± 0.2 cal. ka - Vandergoes et al., 2013) within deformed moraine-bound glaciolacustrine silts in Mangatoetoenui valley (MTO) on Mt. Ruapehu. It is thought that impoundment of the former lake and subsequent deformation of the lacustrine sediments record separate ice advances before and after KOT deposition (Topping, 1974; McArthur and Shepherd, 1990).
On Tongariro massif, well-dated silicic tephras such as Waiohau Tephra (14.0 ka; Lowe et al., 2013) and the Rerewhakaaitu Tephra (17.5 ka; Lowe et al., 2013) have been identified overlying moraine slopes (Topping, 1974; Cronin and Neall, 1997), thus providing minimum age constraints for moraine formation.

Recent applications of surface exposure dating to moraine boulders in TgVC, using cosmogenic $^3$He, have provided the first
direct, quantitative age constraints for late Pleistocene glacier fluctuations in central North Island. On the western side of Tongariro massif, Eaves et al. (2016) show that the innermost moraines in Mangatepopo valley (MPO; Fig. 1) were deposited between 30 - 18 ka, which followed a previous period of glaciation prior to 57 ka. This work confirmed the previous interpretations of an LGM age for these landforms, and supports the wider morphostratigraphic extrapolation of this age inference to proximal, undated moraines (e.g. McArthur and Shepherd, 1990; Barrell, 2011). On the southern slopes of Mt. Ruapehu,
cosmogenic $^3$He surface exposure dating has provided age control for late glacial (15 - 11 ka) moraines, which has further aided the correlation of moraines present outboard of these positions to the LGM (Eaves, 2015). The new ages for moraine formation are also supported recent Ar/Ar dating of ice-bounded lava flows in multiple catchments on Mt. Ruapehu, which shows that valley glaciers were present between 30-20 ka (Gamble et al., 2003; Conway et al., 2015, C.Conway, unpublished data).

# 3 Methods

## 3.1 Glacier model

The aim of this study is to constrain the temperature and precipitation changes from present that may have caused glaciers in central North Island to advance to their LGM extents, using a glacier modelling approach. Several previous studies have
adopted this geologically-constrained glacier modeling approach to reconstruct past climate in New Zealand (e.g. Anderson and Mackintosh, 2006; Golledge et al., 2012; McKinnon et al., 2012; Rowan et al., 2012; Doughty et al., 2013; Putnam et al., 2013b). In this study we use the coupled energy balance - ice flow model of Doughty et al. (2013). A key advantage of this model for our purpose is that mass balance and vertically-integrated ice flow are calculated in two dimensions, which, (1)



captures any potential changes in ice divides that may have occurred under greater ice thicknesses in the past (e.g. Rowan et al., 2012), and (2) produces outputs that are readily comparable to the 2D moraine-based ice mass reconstructions (Fig. 1A). Furthermore, this model is very similar to other models applied in the Southern Alps that have produced quantitative estimates of LGM climate (e.g. Rowan et al., 2012; Putnam et al., 2013a).

### 3.1.1 Mass balance model

To simulate snow accumulation, precipitation is partitioned into rain and snow using a temperature threshold ($T_s$; Table 2). Snow accumulation occurs in grid cells when temperature falls below this threshold, set to $T_s = 0.5\,^{\circ}$ C. The influence of our parameterisations of the energy balance and ice flow models are explored in systematic sensitivity tests (Table 2).

To simulate ablation, the energy balance equation (Eq. 1) is solved within a distributed energy balance model (EBM) as developed (Anderson et al., 2010) and previously applied at individual glaciers and regionally, in contemporary- (Anderson and Mackintosh, 2012) and palaeo-glaciological (Doughty et al., 2013) studies in New Zealand.

$$Q_M = I(1 - \alpha) + L\downarrow + L\uparrow + Q_H + Q_E + Q_G + Q_S + Q_R \qquad (1)$$

where $Q_M$ is the energy available for melt, $I$ is incoming shortwave radiation, $L\downarrow$ is incoming longwave radiation, $L\uparrow$ is outgoing longwave radiation, $Q_H$ and $Q_E$ are sensible and latent heat fluxes respectively, $Q_G$ is the geothermal heat flux, $Q_S$ is the sub-surface heat flux and $Q_R$ is heat input from rain.

Incoming shortwave radiation ($I$) comprises both direct and diffuse components (Oerlemans, 1992). The effect of changing orbital geometry is accounted for using the insolation calculations of Huybers and Eisenman (2006) for 21 ka. Albedo ($\alpha$) is parameterised according to the ELA-dependent scheme of Oerlemans (1992), whereby $\alpha$ increases with elevation and snow thickness, relative to the equilibrium line altitude (ELA = 2483 $\pm$ 50 m asl; Keys, 1988). Following Oerlemans (1992), we use $\alpha_{snow}$=0.72 and explore the impact of this parameterisation in sensitivity tests. Longwave fluxes (L↓ , L↑) include the
effects of surrounding topography, cloudiness and air temperature (Konzelmann et al., 1994; Corripio, 2003; Anderson et al., 2010). Turbulent heat fluxes ($Q_H, Q_E$) are calculated using the bulk method and include the roughness of snow and ice and the Richardson stability criterion (Table 2; Oerlemans, 1992; Klok and Oerlemans, 2002; Anderson et al., 2010).

The sub-surface heat flux ($Q_S$), also commonly referred to as ground heat flux, describes energy exchanges between the
glacier surface and the glacier interior. As we assume that the ice is temperate and that the glacier surface is constantly at the melting point (Oerlemans, 1992), there is no temperature gradient, therefore $Q_S = 0$.





In active volcanic regions, convection and advection of heat to the surface via mantle upwelling and redistribution can raise the geothermal heat flux by several orders of magnitude. In glacierised, active volcanic regions, geothermal heat fluxes ($Q_G$) on the order of $10^0$ - $10^5$ W m$^{-2}$ have been inferred from glacier calorimetry (Clarke et al., 1989; Cuffey and Paterson, 2010). If sustained, such heat fluxes can have a non-trivial impact on glacier mass balance, however these extreme cases are typically only sustained over the order of days to weeks (e.g. Gudmundsson et al., 1997). Thus, while raised geothermal heat fluxes therefore can potentially complicate the climatic interpretation of glacier fluctuations over annual- to decadal timescales (Rivera et al., 2012; Rivera and Bown, 2013), climate is the dominant driver of glacier mass balance and length changes on active volcanoes over centennial to millennial timescales (e.g. Kirkbride and Dugmore, 2001; Mackintosh et al., 2002; Blard et al., 2007; Licciardi et al., 2007; Osborn et al., 2012). We employ a nominal geothermal flux of 1 W m$^{-2}$ ( c. 10 cm water equivalent annual melt) and discuss the possible implications of past volcanism for the palaeoclimatic interpretations derived from glacier model experiments.

We do not include the effects of surface debris cover in our simulations. Historically, debris cover on the glaciers situated on Mt. Ruapehu has varied greatly in space and time. During the most recent sustained volcanic eruptions (AD 1995-96), all glaciers became covered by volcanic products. However, this is quickly removed via surface runoff or buried by winter snow, and today only ice bodies with a low surface slope and those situated close to the current volcanic vent, such as the Summit Plateau and the upper Whangaehu Glacier, remain partially debris-covered. Thus, we acknowledge the potential for past debris cover as an additional source of uncertainty in the simulations and provide further consideration of the specific impacts that these phenomena may have for our findings (Sect. 5).

### 3.1.2 Ice flow model

Ice flow is described using a vertically-integrated, two-dimensional (2D) model based on the shallow ice approximation (SIA) (Plummer and Phillips, 2003; Kessler et al., 2006). A full description is given in Doughty et al. (2013). This formulation assumes ice flow is driven by vertical shear stresses, therefore compressional and tensional (longitudinal) stresses are neglected (Hutter, 1983). We consider that the role of longitudinal stresses on past glacial flow in the glacial troughs studied here would be low, owing to the low-angle bed slopes and absence of steep, bounding valley sides that characterise typical alpine glacier environments. Furthermore, several comparison studies between SIA and higher order ice flow models show little difference in steady-state ice geometries (e.g. Le Meur et al., 2004; Leysinger Vieli and Gudmundsson, 2004), such as those used here for palaeoclimate reconstruction. Thus, the SIA is commonly applied in mountain glacier environments for palaeoclimatic reconstructions, where mass balance imparts greatest uncertainty (e.g. Plummer and Phillips, 2003; Kessler et al., 2006; Doughty et al., 2013).



### 3.1.3 Model input data

Terrain elevation data comes from the New Zealand School of Surveying Digital Elevation Model (NZSoSDEM) (Columbus et al., 2011) and is resampled to 100 m resolution. An ice mask is created using the 'snow/ice' data from the Land Information New Zealand NZMS260 map series. This mask is assigned the mean ice thickness values based on the survey of Keys (1988) and is used to subtract contemporary ice masses from the DEM.

Climate data for the energy balance and snow accumulation models comes from several different sources. Solar radiation and relative humidity are from the Virtual Climate Station Network gridded datasets, sourced from NIWA CliFlo Database (NIWA, 2014), and resampled to 100 m resolution using bilinear interpolation. Following Anderson and Mackintosh (2012), wind speed data comes from the National Centers for Environmental Prediction (NCEP) 850 hPa level, reanalysis data (1981-2010; Kalnay et al., 1996). This dataset is scaled against observational data and applied uniformly over the model domain.

We use 30 year (1981-2010), monthly-mean temperature and precipitation rate data from individual climate stations (NIWA, 2014) distributed around and within the model domain. These data are interpolated onto grids using the methods described in Anderson and Mackintosh (2012) and Doughty et al. (2013). For the temperature interpolations, we use the upland (> 300 m) seasonal lapse rates of Norton (1985), which has been shown to best predict temperatures in alpine regions (Tait and Macara, 2014). During model runs, temperature-dependent components are run at a daily timestep and stochastic variability is introduced to the monthly temperature data via random selection of a normally distributed perturbation value with a mean of zero and a standard deviation of 2.5 °C (*sensu* Golledge et al., 2012). For precipitation grids, we use the mean annual precipitation surface of Tait et al. (2006) to guide the interpolation of station data (see Anderson and Mackintosh, 2012).

### 3.2 Model assessment

As a test of model performance, Fig. 2 compares the modelled, steady state ice extent and mass balance on Mt. Ruapehu summit for present day (i.e. $\Delta T = 0$ and $\Delta P = 0$), against mapped ice extent from Land Information New Zealand topographic maps. Modelled ice extent shows reasonable agreement with glaciers in the majority of the catchments of interest for this study (e.g. Mangatoetoenui, Wahianoa, Mangatururu). However, there is some overprediction of ice in other catchments, particularly for Whangaehu Glacier and Whakapapa Glacier.

Ice accumulation in the Crater Lake basin represents an important source of the excess model ice, particularly for the Whangaehu Glacier, but also Whakapapa Glacier. Today, Crater Lake is a large body of water (c. $10^6$ m$^3$) that occupies the active volcanic vent and fluctuates in temperature between c. 10 - 60 °C over periods of 4 - 16 months (Christenson et al., 2010). Thus, the lake represents an important energy source that prevents snow accumulation at the lake site and raises air temperatures and humidity in the vicinity, thus melting ice flowing towards the lake (Hirst et al., 2012). It is notable that modelled ice extent in catchments that do not receive ice from Crater Lake (i.e. Mangaehuehu, Wahianoa, Mangatoetoenui) are better aligned with





modern observations (Fig. 2).

Based on the increased frequency of hydrovolcanic products in tephra sequences on the surrounding ringplain, Crater Lake is believed to have formed in the mid to late Holocene (c. 3 ka; Donoghue et al., 1997). This post-dates the time period of inter-

est for our glacier modelling application. Thus, although the lake currently represents an important energy source controlling snow accumulation patterns, it is not possible to constrain this for the past, therefore we do not include this energy source in our palaeo-simulations.

A further source of uncertainty in simulating the small contemporary glaciers (e.g. Fig. 2) is the role of wind in redistribut-

ing snow on the summit region. Previous contemporary mass balance observations on Whakapapa and Whangaehu glaciers have noted 'inverted snowlines' (Krenek, 1959; Paulin, 2008), whereby winter accumulation has been redistributed by wind, from the glacial accumulation areas to the lower portions of the glaciers. Including a numerical scheme for this process is computationally expensive (Liston and Sturm, 1998), and has high uncertainties in topographically complex regions such as Mt. Ruapehu, where both the present and past wind fields are largely unknown. Furthermore, wind redistribution of snow is of

greater importance for the mass balance of small glaciers (Kuhn, 1995), such as the contemporary glaciers on Mt. Ruapehu, but less important for larger glaciers, such as those during the LGM, where climatic gradients dominate the mass balance profile. We note that the moraine distribution does not indicate any notable asymmetry in the past ice geometry as might be expected if snow redistribution by prevailing winds was an important control on mass balance in the geological past (e.g. Mitchell, 1996). Thus, we do not attempt to model this process but acknowledge it as a potential source of uncertainty in our mass balance

simulations.

### 3.3 Experimental design

#### 3.3.1 Experiment 1: Step-coolings

Step-coolings from present ($\Delta T$) of -2 to -7 °C, at intervals of 1 °C, are applied uniformly across the domain and the resul-

tant ice masses are allowed to evolve to a steady-state geometry. Equilibrium is achieved when the rate of ice volume change becomes close to zero, which takes 200-350 model years depending on the magnitude of $\Delta T$. These experiments permit an initial assessment of the patterns of ice growth across both volcanoes and the results serve as a guide for the refined catchment-specific simulations of the LGM ice masses carried out in Experiments 2 and 3 (described below).

#### 3.3.2 Experiment 2: Moraine-based simulations

Experiment 2 uses steady-state simulations to constrain the combinations of $\Delta T$ and $\Delta P$ that produce LGM ice extents indicated by the geological evidence in individual catchments (Fig. 1A). Simulations are run at the whole mountain scale in





order to capture any changing ice divides that result from a growing ice mass. For efficiency, separate simulations are run for each volcano. A satisfactory result was obtained when the modelled glacier terminus reached within 1 grid cell (100 m) of the downstream limit of the LGM ice mass, as inferred from geomorphology mapping (above). These simulations are repeated for three precipitation scenarios: +25 %, -25 % and 0 % change from present.

### 3.3.3   Experiment 3: Palaeo-topography

Here we repeat Experiment 2 using modified topographic boundary conditions in order to assess the sensitivity of $\Delta T$ to known post-glacial changes in local topography. A digital elevation model of the pre-15 ka topography has been reconstructed using the results of a 5-year project to constrain the spatial and temporal geomorphic evolution of this region. For example, compi-

lation of existing mapping (e.g. Gamble et al., 2003; Townsend et al., 2008; Price et al., 2012), together with extensive field mapping and new radiometric dating has been used to produce 1:60000 scale maps of the volcanic geology, which delineate individual lava flows (Townsend et al., 2016). To generate palaeo-topographies for these simulations, modern DEMs were manually altered to approximate the post-glacial ($> 15$ ka) topography.

Figure 3 shows the difference between the present day topography and our pre-15 ka surfaces. On Mt. Ruapehu, post-glacial lava flows are concentrated on the northern slopes and extend several kilometeres from the summit region (Fig. 3a). Post-glacial ($<15$ ka) eruption ages have been assigned to these lava flows based on (i) their position within moraine-bound glaciated valleys, (ii) the good preservation of flow structures on the lava surfaces indicating that they have not been subsequently overrun by ice (Hackett and Houghton, 1989), and (iii) recent $^{40}$Ar/$^{39}$Ar dating of several flows within these packages that yield $<$

15 ka ages (C.Conway, unpublished data). In addition, small increases in elevation relative to present have been made on the upper mountain in areas where known post-glacial debris avalanches have occurred (e.g. Palmer and Neall, 1989).

On Tongariro massif, the post-glacial cones of Mt. Ngauruhoe (site 8; $< 6$ ka - Moebis et al., 2011) and North Crater (site 7), and the valley-filling lava flows of Oturere valley (site 10) have been removed (Fig. 3b). To the south of the mountain, the

early-Holocene explosion craters (Topping, 1973; Hackett and Houghton, 1989) now occupied by Tama Lakes (site 9) have been infilled, which represents the only region of significant elevation gain on Tongariro massif.

## 4   Results

### 4.1   Experiment 1: Step-coolings

Figure 4 shows steady-state ice thickness results of Experiment 1, conducted over a domain covering both Mt. Ruapehu and Tongariro massif. Also shown are the inferred LGM ice limits of greater and lesser confidence (Barrell, 2011, modified based



on Eaves et al. (2016)). Modest coolings of 2 - 3 °C from present are sufficient to produce a small ice mass on Mt. Ruapehu, but this temperature change is insufficient to promote significant ice accumulation on Tongariro massif. A cooling of 4 °C is sufficient to meet the well-defined LGM limits in the WAH catchment on southeast Ruapehu, however, the termini of other valley glaciers on this volcano remain well upstream of their mapped limits. In this scenario, ice accumulation on Tongariro is restricted to elevations > c. 1900 m asl, therefore remains well short of the mapped LGM limits. Modelled ice extent approaches the LGM limits in the remaining three catchments (MTO, WHA, MTU) on Mt. Ruapehu when $\Delta T$ = -5 °C and these limits are exceeded in all catchments at $\Delta T$ = -6 °C. On Tongariro massif, modelled ice extent approaches the moraine limits in several catchments (MPO, MHO, UNK) in response to a cooling of 6 °C. At $\Delta T$ = -7 °C, the individual ice masses of the two volcanoes have merged and ice exceeds the LGM limits in all catchments.

## 4.2 Experiment 2: Moraine-based simulations

The precise $\Delta T$ required to simulate the LGM ice geometries delineated by geological evidence in each catchment described above, range from -4.0 to -6.8 °C when precipitation remains unchanged from present (Fig. 5a). Steady state equilibrium line altitudes for these simulations range from c. 1380 - 1660 m asl, which represent depressions of c. 820 - 1100 m from present (Table 3). Imposing a 25 % increase in modern precipitation reduces $\Delta T$ by c. 0.6 °C for all catchments (Fig. 5), meanwhile, decreasing modern precipitation by 25 % requires increases in $\Delta T$ of c. 0.8 °C (Fig. 5a).

Sensitivity tests of key energy balance parameters impact the palaeotemperature reconstructions by up to ±0.5 °C for the chosen ranges (Fig. 5b). Altering the albedo of snow ($\alpha_{snow}$ = 0.67 - 0.77) and snow-temperature threshold ($T_{snow}$ = 0 - 1.5 °C) have the greatest effects (c. ± 0.1 - 0.5 °C). Using a temperature lapse rate (d$T$/d$z$) of -0.006 °C m$^{-1}$, uniformly applied across all months, decreases $\Delta T$ by 0.1 - 0.2 °C. Changing the characteristic ice roughness length ($Z_{ice}$ = 0.0008 - 0.01 m) also causes deviations in $\Delta T$ of ± 0.1 - 0.2 °C, relative to the optimal setting. Flow parameters $U_c$ and $A$ have negligible (<0.1 °C) impact on $\Delta T$.

Findings from Experiment 1 highlighted the variability in the LGM palaeoclimate reconstructions that exists, both between volcanoes, and between individual catchments. Figure 5a shows that the Wahianoa (WAH) catchment on Mt. Ruapehu requires a conspicuously lower amount of cooling to match the identified LGM ice limits ($\Delta T$ = -4.0 °C, when $\Delta P$ = 0), compared to all other catchments studied (c. -5.2 to -6.8 °C, when $\Delta P$ = 0). Also, there is an offset in the temperature forcings necessary to simulate the mapped LGM ice limits between the two volcanoes. Catchments on Tongariro massif range require coolings of c. -6.0 to -6.8 °C (mean = -6.3 °C), when precipitation is unchanged, whilst catchments on Mt. Ruapehu require -4.0 to -5.8 °C (mean = 5.0 °C, or 5.4 °C when WAH is removed). Finally, the results presented in Fig. 5 represent the climatic forcing, from present, required to meet the inferred downstream limits of LGM glaciation. In several catchments, ice spills over ice-marginal indicators, such as lateral moraines, before the geologically-constrained termini are reached (Fig. 6b,c). The possible reasons



for the discrepancies, and the potential implications for palaeoclimate estimates are discussed below (Sect. 5.1).

### 4.3 Experiment 3: Palaeo-topography

Figure 7a shows the change in temperature from present required to simulate the mapped LGM geometries in the nine catchments, using topographic boundary conditions that approximate that of the LGM (Fig. 3). When precipitation remains unchanged from present, $\Delta T$ ranges from c. -4.1 to -7.1 °C, which represent differences of +0.1 to -0.5 °C from Experiment 2 (Fig. 7b). In all catchments except one (MTO), more cooling was required, relative to Experiment 2 (Fig. 7b). This is likely because the majority of the imposed topographic changes involved subtraction of depositional units (Fig. 3), which increased local surface air temperature in the model. The greatest changes in $\Delta T$ between Experiment 2 and Experiment 3 occurs in the UNK catchment, where c. 0.5 °C of extra cooling is required to simulate the inferred glacial geometries (Fig. 7b). This change is the result of reduced ice flux from the vicinity of Mt. Ngauruhoe, caused by the elevation reduction in the accumulation area, following removal of this Holocene-aged volcanic cone (Fig. 3). However, this topographic alteration has less of an impact in the WAI catchment, where $\Delta T$ is reduced by 0.2 °C, relative to Experiment 2. Whilst removal of Mt. Ngauruhoe may act to channel ice flow into the WAI catchment, the overall reduction in elevation reduces snow accumulation, therefore reducing the impact on $\Delta T$. The imposed topographic changes did not improve the poor fit between modelled ice geometry and the geological constraints, as ice still spills over lateral moraines in the WAI and MTA catchments before reaching the inferred LGM termini.

On Mt. Ruapehu, the major changes to the topography were made in the MTO catchment, which is the only catchment where $\Delta T$ decreased by c. 0.1 - 0.2 °C, relative to Experiment 2 (Fig. 7b). In this instance, the removal of syn- and post-LGM lava flows in the upper and middle parts of this catchments has resulted in increased ice flux to the lower valley, despite the overall reduction in elevation. This is caused by a reduction in the ice flux leaving the catchment through overspill, which helps offset the effect of increased temperature caused by the reduction in bed elevation. In the other catchments on Mt. Ruapehu (MTU, WAH, WHA), $\Delta T$ was reduced by 0.1 - 0.2 °C relative to Experiment 2. Thus accounting for post-glacial changes in bed topography cannot resolve the anomalous $\Delta T$ result in the WAH catchment. Finally, there remains a poor fit between the geologically-inferred lateral ice margins in the MTU catchment and those simulated in Experiment 3.

## 5 Discussion

Using a numerical glacier model, we have provided quantitative constraints of the temperature depression from present in central North Island during the LGM and the sensitivity of these results to model parameterisation and changing topographic boundary conditions. The main findings are as follows: (1) the temperature depression required to simulate the LGM glacial geometries of individual catchments varies between -4.0 and -6.8 °C; (2) there is a systematic offset of c. 1 °C in the model-





derived palaeotemperatures associated with LGM moraines between the two volcanoes; (3) using geologically-constrained reconstructions of LGM topography has relatively little impact (+0.2 to -0.5 °C) on the palaeotemperature reconstruction. Below, we first consider the possible sources of uncertainty, before placing the results in context of other, local terrestrial and marine palaeotemperature proxy reconstructions.

## 5.1 The role of changing topography on palaeoclimate estimates

Improved constraint of the timing and extent of late Quaternary volcanism in central North Island (e.g. Gamble et al., 2003; Conway et al., 2015, and unpublished data) has allowed a test of the impact that changing topographic boundary conditions have on palaeoclimate estimates for glacier modelling. Using expert-defined topographic reconstructions, informed by recent

field mapping and radiometric dating, the temperature forcing required to simulate the inferred LGM glaciers is altered by +0.2 to -0.5 °C. The majority of the imposed topographic changes serve to remove post-glacial lava flows that have built volcanic cones (e.g. Mt. Ngauruhoe) or infilled glacial troughs (e.g. MTO, MPO; Fig. 3). Subtraction of these features from the contemporary land surface has lowered the glacier bed elevation, which raises the local surface air temperature and explains why most catchments require increased cooling to achieve the LGM limits in Experiment 3, relative to Experiment 2. This result

mirrors the findings of McKinnon et al. (2012), who found that subtraction of post-glacial sedimentary fill from the Pukaki basin, in central Southern Alps, resulted in lower glacier model derived palaeotemperature estimates, relative to studies that used present day bed topography.

Removal of the post-glacial lava flows in the vicinity of the MTO catchment reduced the temperature forcing required to

simulate the LGM ice geometry in that catchment by c. 0.2°C, relative to Experiment 2. Reduced overspill, as shown by the improved fit between the model output and the lateral moraines, indicates that the retention of ice within the MTO catchment was improved by the imposed topographic changes and this effect was sufficient to offset the decreased accumulation / increased ablation induced by land surface lowering. However, the imposed topographic changes did not improve the fit in all catchments where overspill occurs (e.g. MTA, WAI, MTU), thus it is probable that the LGM palaeotemperature estimates

presented in Fig. 5 and Fig. 7 for these catchments overestimate the true magnitude of temperature lowering associated with the LGM moraines in these catchments. This interpretation is supported by the fact that these catchments require the greatest magnitude of cooling, relative to present, of all catchments on the respective volcanoes. Discounting the reconstructions from these catchments leaves LGM palaeotemperature estimate ranges of -4.1 to -5.4 °C for Mt. Ruapehu and -6.1 to -6.3 °C for Tongariro massif (ΔP = 0; Experiment 3). Thus, accounting for ice overspill and known topographic changes is insufficient to

resolve the apparent offset between LGM palaeotempertaure estimates between these two volcanoes.

Geometric constraint of well-preserved post-glacial lava flows can be achieved with relative ease via detailed field investigations, however the recognition of post-glacial erosional events (e.g. sector collapse, fluvial incision) and subsequent topographic reconstruction is less straightforward. Some erosional events are identifiable in the modern landscape on Mt. Ruapehu





(e.g. Murimotu Formation sector collapse at 10.4 - 10.6 cal. ka BP; Eaves et al., 2015), however the precise source locations and pre-erosional topographies remain highly uncertain (e.g. Hackett and Houghton, 1989; Palmer and Neall, 1989; McClelland and Erwin, 2003). Such erosional events alter drainage pathways and bed hypsometries, with potential implications for modelled ice distributions and palaeoclimatic reconstruction. For example, it can be hypothesised that the offset in LGM temperature reconstructions between the glacial catchments of Mt. Ruapehu and Tongariro massif is caused by post-glacial change in the relative altitudes of the two volcanoes. A post-glacial decrease in the summit altitude of Tongariro massif, relative to Mt. Ruapehu, could explain the need for greater cooling on Tongariro massif in the simulations presented here. However, there is little geological evidence to support the notion that Tongariro massif has experienced major post-glacial degradation, nor that Mt. Ruapehu has significantly increased in elevation since the LGM. This absence of evidence, combined with the fact that known changes in topographic boundary conditions had relatively little impact on palaeotemperature reconstructions (Experiment 3), makes it unlikely that post-glacial topographic change is the primary source of this systematic offset.

### 5.1.1 Other sources of uncertainty

The glacier modelling experiments presented here suggest that stadial conditions in central North Island were characterised by temperatures 4 to 7 °C lower than present (Fig. 5). Here we first discuss the possible reasons for the inter-catchment and inter-volcano variability in palaeotemperature reconstructions, before placing our results in the context of other, quantitative palaeoclimate reconstructions from the New Zealand region.

Our results show a c. 3 °C spread in palaeotemperature estimates between individual catchments and a c. 1 ° C disparity between the two volcanoes. Given the relatively small influence of known topographic changes as a driver, we consider these differences can be explained predominantly by the combination of three principal factors, which we discuss below.

First, differences in climate forcing may arise from chronological uncertainty. In only two of our studied catchments (MPO, WAH) have the moraines targeted in our model experiments been shown to pertain to the LGM (e.g. Eaves et al., 2016; Eaves, 2015). Moraines representing glacial advances prior to the LGM have been recognised and dated on Tongariro massif (Eaves et al., 2016), therefore it is possible that some of the limits we targeted predate the LGM. However, Eaves et al. (2016) noted a distinct morphological difference in the preservation of moraines formed early in the last glacial cycle (> 59 ka) and those constructed during the LGM. The undated moraines targeted in our study had been correlated to the LGM based primarily on their similar morphology and degree of preservation to dated landforms, therefore we consider it unlikely that our simulations have targeted pre-LGM glacial limits. A separate possibility is that our targets represent different advances within the LGM. This cold climatic interval spans c. 10 ka (28 - 18 ka), during which time there were significant fluctuations in climate over millenial-timescales (Barrell et al., 2013). Indeed, exposure ages of LGM moraines do show evidence for multiple moraine building episodes 30 - 18 ka (e.g. Eaves et al., 2016). Thus, the inter-catchment variability seen in our palaeotemperature estimates may in part reflect the range of temperature cooling events during this prolonged but variable cold period.



Second, inaccuracies in the present day climate data, particularly precipitation distribution, may impart some error to the palaeotemperature reconstructions. The paucity of high-altitude precipitation data for the present day represents an important source of uncertainty in both contemporary and palaeo applications of glacier mass balance models. For example, Rowan et al.

(2014) find that uncertainty in present day precipitation distribution imparts uncertainty of up to 25 % in modelled LGM glacier length in the central Southern Alps, which equates to about 0.5 °C in the palaeotemperature estimate. The good agreement between observed ice distribution on Mt. Ruapehu and ice geometries simulated using the 30-year (AD 1981-2010) average climate datasets (Fig. 2) provides confidence that these datasets provide a useful starting point from which to assess the local LGM climate anomaly in catchments on Mt. Ruapehu. However, no glaciers exist on Tongariro massif today and there are no

climate stations on the volcano, therefore it is more difficult to assess how representative the modern climate grids are for this volcano.

Underestimation of the contemporary precipitation rate on Tongariro massif provides one possible explanation for our finding that slightly greater temperature forcings are needed to match the LGM limits on Tongariro massif, relative to Mt. Ruapehu.

The precipitation-temperature relationships presented in Fig. 5a indicate that precipitation changes of ± 25 % are balanced by temperature changes of ± c. 0.6 - 0.8 °C, which is consistent with similar estimates for South Island glaciers (Oerlemans, 1997; Anderson and Mackintosh, 2012). Thus, precipitation on Tongariro would need to be increased by 30 - 50 %, relative to Mt. Ruapehu, in order to account for the c. 1.0 - 1.3 °C temperature difference associated with the inferred LGM glacial limits between the two volcanoes. This magnitude is quite large and we consider it unlikely that the spatial precipitation gradient

in the contemporary datasets is inaccurate to such a degree, however this may still represent an important contributing factor. Potential past changes in local precipitation gradients, for example arising from atmospheric circulation changes (e.g. Lorrey et al., 2012), may also contribute. Improved constraint of present and past precipitation rates will reduce these uncertainties.

Third, is possible spatial heterogeneities in key glaciological parameters. The sensitivity tests presented in Fig. 5b, provide

a first-order assessment of the uncertainty imparted by parameters in the energy balance model. Varying key parameters within acceptable bounds causes deviations in reconstructed temperatures of up to ±0.5 °C, which indicates that some of the inter-catchment variability in palaeotemperatures (e.g. WAH catchment) could be explained by spatial heterogeneities in model parameters, which are currently assumed uniform across the model domain.

Our sensitivity analyses show that palaeotemperature estimates are most sensitive to albedo (Fig. 5), which is unknown for the pre-historic period. Surficial debris cover lowers the albedo of ice, but can act to enhance or reduce surface melt on glaciers depending on the thickness of the debris layer, which in turn is dictated by sediment availability. Richardson and Brook (2010) measured ice ablation rates of clean and tephra-covered snow on Summit Plateau at Mt. Ruapehu and found that tephra thicknesses of up to 7 cm enhanced ablation rates, relative to that of clean snow. In temperate regions, empirical observations have

shown that tephra deposited on glacier surfaces is quickly redistributed by surface runoff, and thus has a net long-term effect



of enhanced surface ablation following deposition (Kirkbride and Dugmore, 2003). Thus, the presence of thin (cm-scale) debris cover on some LGM glaciers in central North Island would have enhanced ablation and would cause underestimation of palaeotemperatures for these catchments in our model that assumes clean glacier surfaces. This process may contribute to the inter-catchment, and/or inter-volcano variability in our LGM palaeotemperature estimates.

Despite these possible sources of uncertainty, the weight of evidence afforded by our multi-catchment approach allows us to say with reasonable confidence that LGM temperatures in central North Island reached at least 5 °C colder than present. Meanwhile it is unlikely that temperatures were depressed by more than 7 °C relative to today. This is in good agreement with other quantitative LGM temperature reconstructions from New Zealand (e.g. Golledge et al., 2012; Newnham et al., 2013; Seltzer et al., 2015; Fig. 5), which supports our conclusion that climate represents the dominant signal recorded by the Quaternary moraine record on the central North Island volcanoes. In the next section we discuss our finding in the context of other palaeoclimate records.

## 5.2 Last Glacial Maximum climate in New Zealand

The glacier modelling experiments presented here suggest that stadial conditions in central North Island during the LGM were characterised by temperatures 4 to 7 °C lower than present (Fig. 5), although most of the catchments studied require cooling of c. 5.1 - 6.3 °C to achieve the mapped ice limits, when precipitation is unchanged from present. Quantitative estimates of regional changes in precipitation rate during the LGM remain poorly constrained, although evidence from climate modelling (Drost et al., 2007; Rojas et al., 2009), previous glacier modelling (Golledge et al., 2012), carbon isotopes in speleothems (Whittaker et al., 2011) and diatoms in maar deposits (Stephens et al., 2012) indicate that drier than present conditions prevailed across New Zealand at this time. Precipitation reductions of up to 25 % from present require additional decreases in temperature by up to 0.8 °C to achieve the LGM glacial geometries in central North Island (Fig. 5). Such a change in precipitation is likely a maximum estimate given that climate model simulations predict changes in total annual precipitation of < 10 % (e.g. Drost et al., 2007).

Steady-state equilibrium line altitudes of the simulated LGM glaciers fall between c. 1400 - 1650 m asl (Table 3), which represent depressions of c. 800 - 1100 m, relative to present. Despite methodological differences, our estimate using physically-based modelling is in agreement with that of McArthur and Shepherd (1990), who manually reconstructed the LGM ELAs on Mt. Ruapehu to be 1500 - 1600 m asl. Eaves et al. (2016) estimated the ELA of the LGM glacier in the MPO catchment as c. 1400 - 1550 m asl, using the accumulation area ratio (AAR) and maximum elevation of lateral moraine (MELM) methods, which agrees well with the model simulation presented here (c. 1510 m asl; Table 3). This good agreement between model and manual reconstruction of glacier ELAs echoes previous similar findings (e.g. Kaplan et al., 2010; Doughty et al., 2013), demonstrating the utility of simple ELA reconstructions for efficient extraction of palaeoclimate data from moraine records.

Lowering of the regional ELA to 1500 m asl at the LGM is insufficient to promote widespread glaciation in the mountain



ranges elsewhere in North Island, as few other peaks exceed this elevation. The only other existing evidence for LGM glacia-
tion in North Island comes from the Tararua Range in southern North Island, where the local *p*ELA was estimated to be c. 1100
m asl (Brook et al., 2005; Brook et al., 2008). This reconstruction is considerably lower than our reconstruction and recon-
structions from elsewhere in New Zealand (Porter, 1975; McCarthy et al., 2008; Golledge et al., 2012), which may represent
topo-climatic controls on mass balance of this former cirque glacier, such as wind-driven snow accumulation. The absence of
contemporary glaciers in the Tararua Range precludes robust spatial comparison of ELA depressions to the results presented
here. Quantitative palaeotemperature estimates from North Island have been made using fossil pollen assemblages and ground-
water noble gas paleothermometry, which also indicate LGM temperature depressions of 4-7 °C below present (McGlone and
Topping, 1977; Sandiford et al., 2003; Wilmshurst et al., 2007; Newnham et al., 2013; Seltzer et al., 2015), which is consistent
with our glacier model results.

Several glacier-based assessments of LGM temperature have previously been made for South Island, New Zealand, using
a variety of different glacier models. Simulations of the entire Southern Alps icefield, using a degree-day model coupled to the
Parallel Ice Sheet Model, indicate that the LGM was characterised by temperatures 6 - 6.5 °C colder than present, coupled with
a reduction in precipitation of c. 25 % (Golledge et al., 2012). It is notable that the best-estimate palaeotemperature scenarios
did not achieve a good fit between modelled ice extent and the geological evidence in all catchments (Golledge et al., 2012,
their Fig. 10B), which may also reflect some of the uncertainties discussed in Sect. 5.1.1 above. Using a different glacier model
with higher grid resolution and a different representation of modern climate, Rowan et al. (2013) and Putnam et al. (2013a)
achieve a good model fit in one region where Golledge et al. (2012) did not (e.g. Rakaia), despite using a similar temperature
forcing ($\Delta T$ = -6.25 to -6.5 °C). Using the University of Maine Ice Sheet Model, Putnam et al. (2013b) find that a cooling of
$6.25 \pm 0.5$ °C (with no precipitation change) is required to generate an ice extent that matches well-dated moraines in the Lake
Ohau catchment. When precipitation is reduced by 30 % the required cooling increases to 6.9 °C. These studies have shown
that, despite differences in boundary conditions and formulations for glacier flow, glacier model experiments consistently sug-
gest peak stadial air temperatures during the LGM were 6-7 °C cooler in the Southern Alps, which is in good agreement with
our estimates from central North Island.

## 6  Conclusions

1.  Simulations of nine glaciers in central North Island, New Zealand using a 2D, coupled energy-balance, ice-flow model,
suggests that local air temperatures were depressed by 4 - 7 °C relative to present during the Last Glacial Maximum.
    The spread of temperature estimates primarily reflects uncertainties in dating and model parameters, however the weight
    of evidence allows a best estimate to be made of 5.1 - 6.3 °C below present. A decrease in precipitation (as suggested
    by proxy evidence and climate models) of up to 25 % from present, increases the magnitude of the required temperature



changes by up to c. 0.8 °C.

2. Accounting for volcanically-induced, post-glacial topographic change generally decreases the elevation of the glacier bed elevation, which increases the magnitude of cooling required to simulate the former ice limits. The imposed topographic changes do not significantly change past glacial drainage patterns, although the difficulty in reconstructing pre-erosional topographies makes it hard to fully assess this possibility. The impact of topographic change on the temperature reconstructions is variable between catchments, with changes on the order of 0.1 - 0.5 °C, relative to the simulations with present day land surface.

3. Our palaeoclimatic reconstructions are in agreement with proximal temperature reconstructions from pollen (Newnham et al., 2013) and groundwater (Seltzer et al., 2015), as well as several similar model-based estimates from glacial records in central Southern Alps (Golledge et al., 2012; Putnam et al., 2013b; Rowan et al., 2013). This growing body of evidence indicate that air temperatures across most of New Zealand were depressed by c. 6 °C relative to present during the Last Glacial Maximum.

*Acknowledgements.* S.R.E. was supported by the Victoria University Doctoral Scholarship, a VUW Faculty Strategic Research Grant and the Antarctic Research Centre Endowed Development Fund.





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



**Table 1.** Glacial catchments targeted in our model experiments. (*Glacier became extinct post-1988 survey)

| Glacier | Code | Extant glacier (Y/N) | Valley aspect | ELA (m asl; Keys, 1988) |
|---|---|---|---|---|
| **Mt. Ruapehu:** | | | | |
| Whakapapaiti | WHA | N* | NW | 2550-2650* |
| Mangaturuturu | MTU | Y | SW | 2450-2600 |
| Wahianoa | WHA | Y | SE | not reported |
| Mangatoetoenui | MTO | Y | NE | 2400-2500 |
| **Tongariro massif:** | | | | |
| Mangatepopo | MPO | N | W | - |
| Waihohonu | WAI | N | SE | - |
| Mangatawai | MTA | N | E | - |
| Unnamed | UNK | N | E | - |
| Mangahouhounui | MHO | N | E | - |

**Table 2. Optimal glacier model parameter settings, sources and sensitivity test values.**

| Parameter | Value | Source | Sensitivity test values |
|---|---|---|---|
| Snow albedo ($\alpha_{snow}$) | 0.72 | (Oerlemans, 1992) | 0.67 - 0.77 |
| Snow-rain threshold temperature ($T_s$) | 0.5 °C | (Hendrikx and Hreinsson, 2012) | 0 - 1.5 °C |
| Temperature lapse rate ($\frac{dT}{dz}$) | Seasonal | (Norton, 1985) | 6 °C km$^{-1}$ |
| Ice roughness ($z_{ice}$) | 0.004 m | (Anderson and Mackintosh, 2012) | 0.0008 - 0.01 m |
| Snow roughness ($z_{snow}$) | 0.001 m | (Anderson and Mackintosh, 2012) | 0.0005 - 0.002 m |
| Typical sliding velocity ($U_c$) | 50 m yr$^{-1}$ | (Anderson et al., 2015) | 20 - 80 m yr$^{-1}$ |
| Glen's flow law coefficient ($A$) | 2.14 x 10$^{-16}$ Pa$^{-3}$ yr$^{-1}$ | (Paterson, 1994) | 1e$^{-15}$ - 1e$^{-18}$ Pa$^{-3}$ yr$^{-1}$ |



**Table 3. Palaeo-equilibrium line altitudes (pELA) of the simulated LGM glaciers and the difference from present using (dELA$_{mean}$ (m) = arithmetic mean of Keys (1988) = 2483 m asl; dELA$_{obs}$ = change from the mid-point of observed present day ELAs of individual glaciers given by Keys (1988), where available (see Table 1).**

| Catchment | pELA (m asl) | dELA$_{mean}$ (m) | dELAobs | Modelled $\Delta$T ($^{\circ}$C) |
|---|---|---|---|---|
| Mangatawai (MTA) | 1380 | -1103 | - | -6.8 |
| Waihohonu (WAI) | 1390 | -1093 | - | -6.7 |
| Mangahouhounui (MHO) | 1460 | -1023 | - | -6.4 |
| Mangatepopo (MPO) | 1510 | -973 | - | -6.0 |
| Unnamed (UNK) | 1550 | -933 | - | -5.8 |
| Mangaturuturu (MTU) | 1530 | -953 | -995 | -5.7 |
| Mangatoetoenui (MTO) | 1580 | -903 | -870 | -5.3 |
| Whakapapaiti (WHA) | 1550 | -933 | -1050 | -5.2 |
| Wahianoa (WAH) | 1660 | -823 | - | -4.0 |





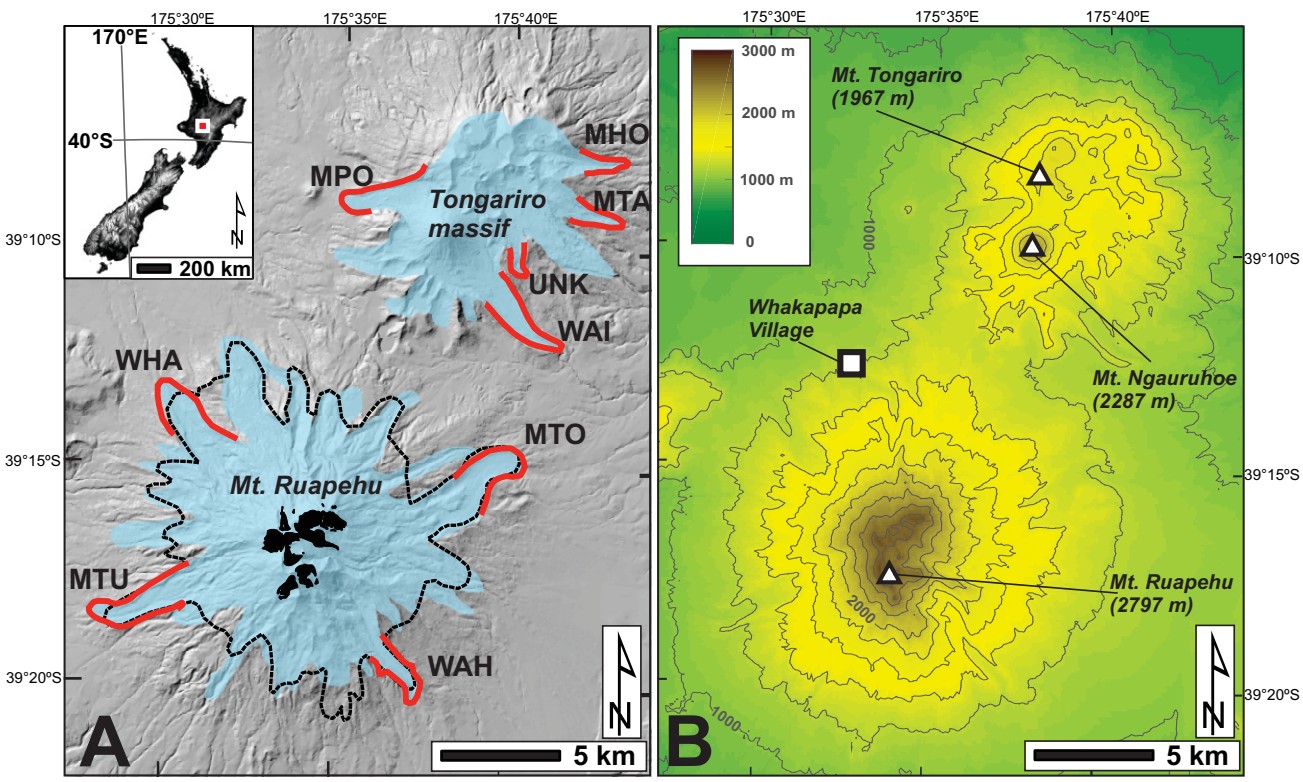

**Figure 1.** A. Present day ice distribution (black polygons) and previous reconstructions of the Last Glacial Cold Period ice masses (Barrell, 2011 - blue polygon; McArthur and Shepherd, 1990 - black dashed line) on the central North Island volcanoes: Tongariro massif and Mt. Ruapehu. Red lines denote LGM ice limits targeted in Experiments 2 and 3 of this study, for the following valleys listed in Table 1. Inset map shows location of Tongariro Volcanic Centre in central North Island, New Zealand.; B. Topography and selected peaks (Land Information New Zealand - LINZ)





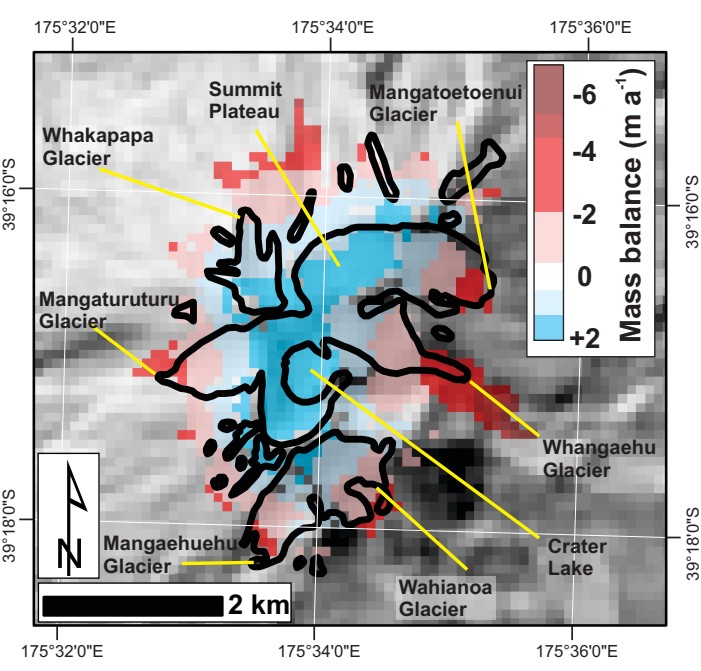

**Figure 2.** Modelled steady state glacier extent and mass balance on Mt. Ruapehu summit, compared to the mapped extent of ice and perennial snow from Land Information New Zealand (black outlines). Prominent glaciers and landforms mentioned in text are labelled.





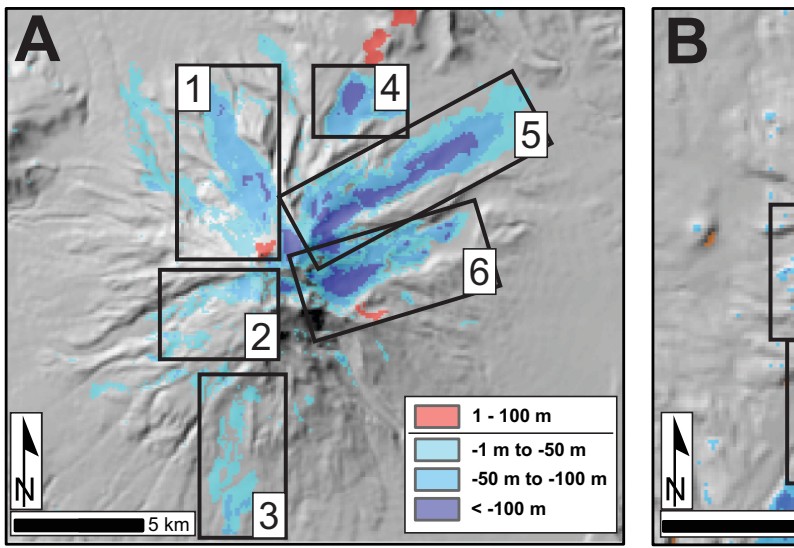
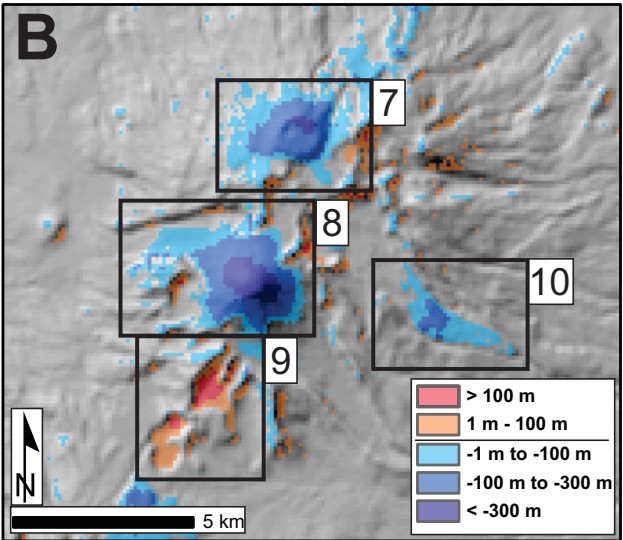

**Figure 3.** Geologically informed reconstruction of pre-15 ka topography, displayed as a difference (in metres) from modern topographic boundary conditions, for glacier model domains of (a) Mt. Ruapehu (1 = Whakapapa flows (Hackett and Houghton, 1989); 2 = Holocene flows in upper Mangaturuturu (C.Conway, unpublished Ar/Ar data); 3=Rangatauanui flow; 4 = Saddle Cone flow(s); 5-6 = Post LGM flows in and adjacent to Mangatoetoenui valley; (Gamble et al., 2003)), and (b) Tongariro massif (7 = North Crater; 8 = Mt. Ngauruhoe; 9 = Tama Lakes explosion craters; 10 = Oturere flows). Blues = surface lowering relative to present (negative change); reds = elevation gain relative to present (positive change).





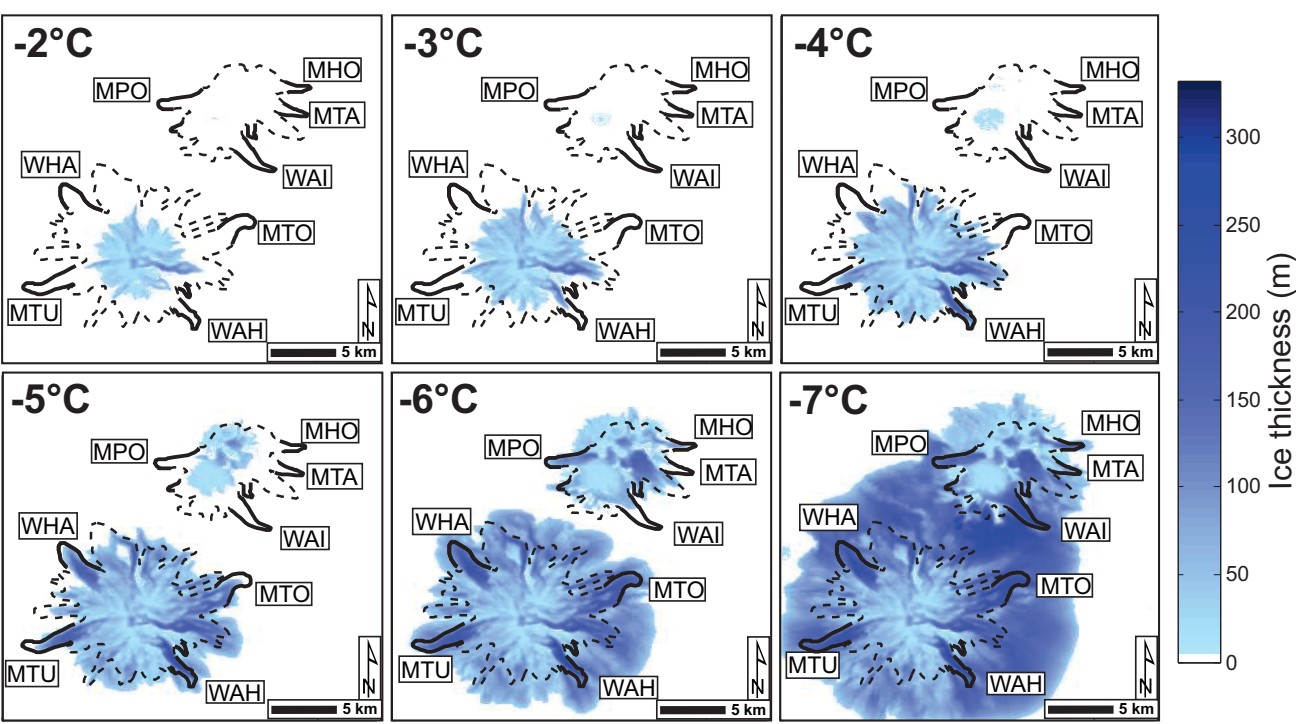

**Figure 4.** Experiment 1: Modelled, steady-state ice thickness (in metres) and extent on Mt. Ruapehu and Tongariro massif resulting from step coolings of -2 °C to -7 °C from present, with precipitation unchanged. Solid and dashed lines represent geologically constrained LGM ice limits of greater and lesser confidence, respectively (modified from Barrell, 2011). See text for catchment labels.





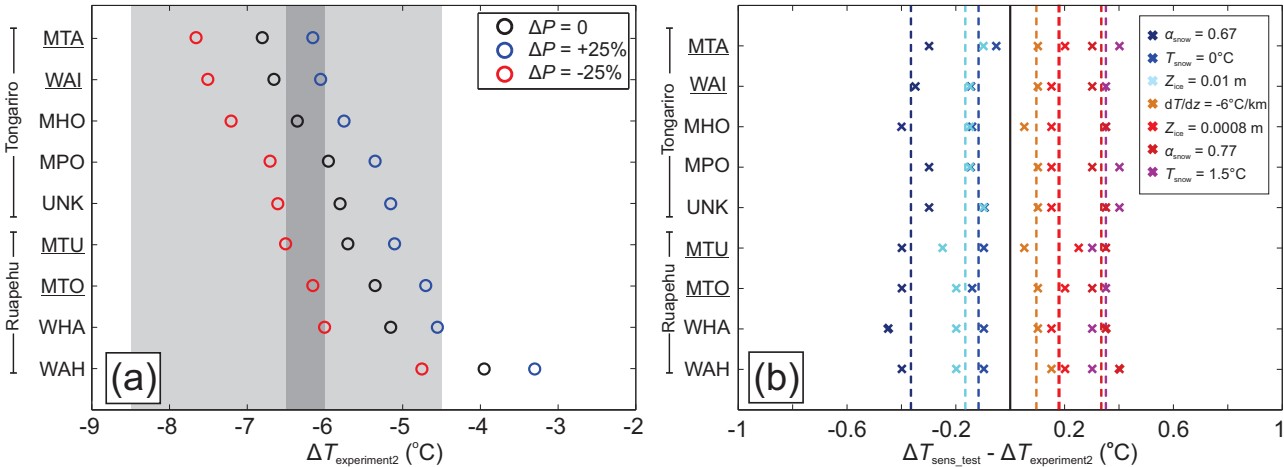

**Figure 5.** Experiment 2: (a) Temperature forcing, from present, necessary to simulate inferred LGM ice geometries in catchments on Mt. Ruapehu and Tongariro massif. Results shown for three precipitation change ($\Delta$P) scenarios: 0 % (black); +25 % (blue); and -25 % (red) change from present. Underlined labels on the y-axis represent catchment where modelled ice thickness spills over ice-marginal landforms, before reaching the geologically inferred terminus (see text). Grey shading depicts the pollen-based southern North Island LGM temperature lowering estimate of Newnham et al. (2013) (-6.5 $\pm$ 2.0 °C) - derived using the partial least squares method and the glacier model derived Southern Alps LGM cooling estimate of Golledge et al. (2012) (-6.0 to -6.5 °C, when precipitation is reduced by 25 %). (b) The sensitivity of palaeotemperature reconstructions for the following parameters: albedo of snow ($\alpha_{snow}$), snow temperature threshold ($T_{snow}$), air temperature lapse rate (d$T$/d$z$) and ice surface roughness ($Z_{ice}$). Dashed lines indicate the mean impact of each sensitivity test. Flow parameters $U_c$ and $A$ (not shown) have negligible (<0.1 °C) impact on $\Delta T$.




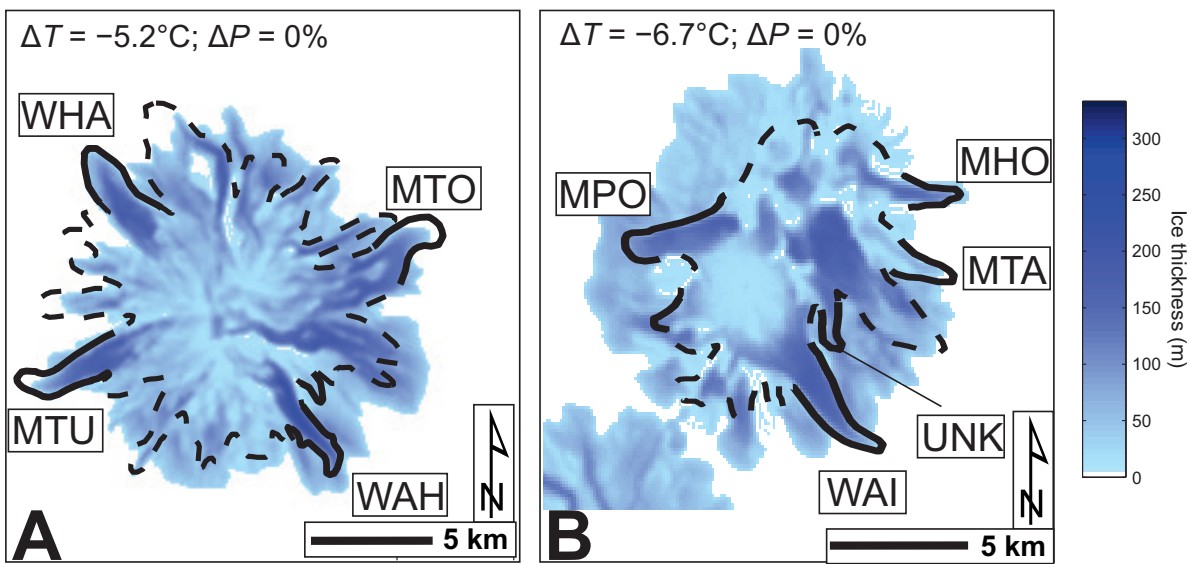

**Figure 6.** (a) Modelled, steady-state ice geometry on Mt. Ruapehu when $\Delta T$ = -5.2 °C and $\Delta P$ = 0 (Experiment 2). This represents the best-fit simulation for the WHA catchment; (b) Steady state ice geometry on Tongariro massif when $\Delta T$ = -6.7 °C and $\Delta P$ = 0 (Experiment 2). This is the best fit to the inferred LGM terminus in the WAI catchment, however note the ice overspill at the lateral margins.




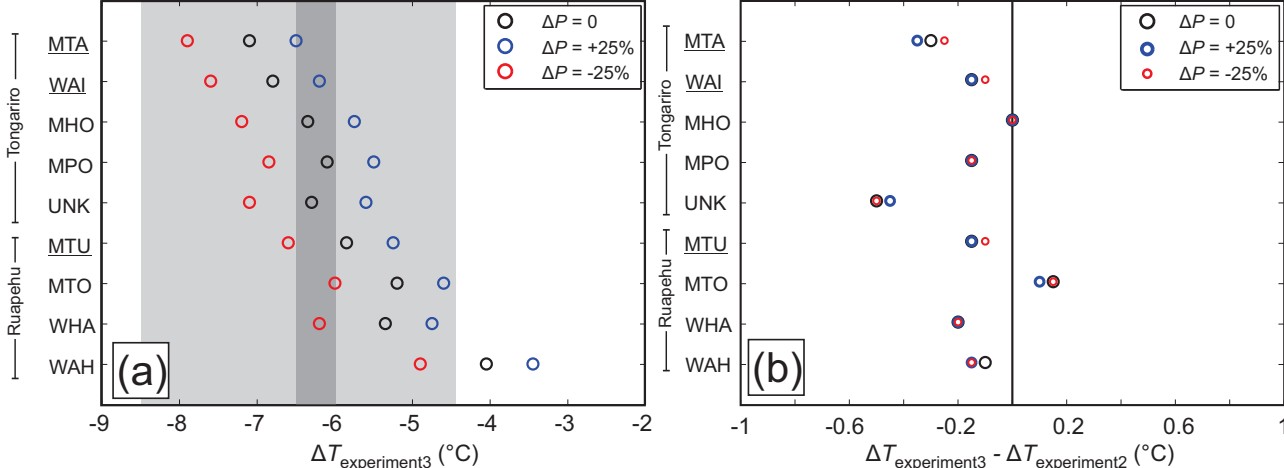

**Figure 7.** Experiment 3: (a) Temperature forcing, from present ($\Delta T_{experiment3}$), necessary to simulate inferred LGM ice geometries in catchments on Mt. Ruapehu and Tongariro massif, using the modified, pre-15 ka topography. Results shown for three precipitation change ($\Delta P$) scenarios: 0% (black); +25% (blue); and -25% (red) change from present. Underlined labels on the y-axis represent catchments where modelled ice thickness spills over ice-marginal landforms, before reaching the geologically-inferred terminus (see text). Vertical black lines and light grey shading depict pollen-based southern North Island LGM temperature estimate of Newnham et al. (2013) (6.5 ± 2.0 °C) - derived using the partial least squares method. Dashed black lines and dark grey shading depict the glacier model derived Southern Alps LGM temperature estimate of Golledge et al. (2012) (6.0 - 6.5 °C, when precipitation is reduced by 25%). (b) The difference in $\Delta T$ between model simulations in Experiment 3 ($\Delta T_{experiment3}$) and Experiment 2 ($\Delta T_{experiment2}$).