# Peer review of "The Last Glacial Maximum in central North Island, New Zealand: palaeoclimate inferences from glacier modelling"

_Climate of the Past, 2016_

## Referee Comment (RC1) · Ann V Rowan (Referee) · 2 Feb 2016

The authors present a glacier modelling study that reconstructs the LGM advance of glacier tongues from small ice caps on the summits of two volcanic peaks in the North Island of New Zealand. New Zealand is a key site for exploring the drivers and geomorphological impact of past climate change, and the North Island is relatively understudied compared to the South Island due to the much less extensive glaciation in the North, making this manuscript novel in attempting a 2-D model-based reconstruction of glaciers in northern New Zealand. These small, marginal glaciers (some of those under investigation have vanished since the last glacial) can provide important information about rapid climatic variations, which adds further value to this study. The main

limitation of the manuscript is that the glacier model simulations only investigate conditions approximate to the Last Glacial Maximum (LGM), rather than the present day or postglacial. A good test of this sort of modelling is if the study glaciers can be simulated under a range of known conditions. Although the authors may have performed present day simulations, and have data describing post-LGM glacier limits, the results are not presented. The manuscript is generally well written, although there are some minor inconsistencies in the text and some editing needed. The figures are clear and well presented although the manuscript would benefit from containing more visual information than is the case at present to communicate the simulations and results more clearly.

Main points

Model comparison with present day glaciers. The glacier model is tuned to LGM extents and the present-day simulated mass balance is shown in Fig. 2. However, a simulation of present day ice extents is not provided, which would give a much more valid comparison with the mapped ice extents presented in Fig. 2 as it is difficult for the reader to imagine the extents generated by the presented mass balance. Tuning a glacier model to a single snapshot of the glacial history is relatively easy compared to simulating multiple phases of glaciation to demonstrate that the model parameterisation is valid. The authors should present a simulation under present-day climate conditions, and if there is geomorphological evidence for glacier extents post-LGM then these simulations could also be presented even if their ages are unknown, which may be valuable to subsequent research projects.

Geomorphological evidence for glacier advance. The moraine limits presented in Fig. 1 are derived from field mapping by several studies including the thesis work of the current author. The manuscript would benefit from a more detailed, maybe separate, first figure presenting the different landforms mapped by each study and identifying landscape modification by post-LGM volcanism, to convince the reader that the geomorphological evidence used to constrain the modelling does represent (at least fairly)

[Figure]

stable LGM glacier termini. This could also clarify what was the basis for choosing the ice margins to fit the model to, as indicated by the red lines in Figure 1. Also, a useful point to add to the aims of the paper (P2) and address; is there evidence for post-LGM moraine formation, or is it likely that this would have been removed by volcanism? Does the difference in simulated dT for each volcanic peak result from the moraines in each location representing different stages of the LGM, or did each peak show a different response to LGM cooling?

Description of the glacier model. The description of the model set up (Section 3.1.1) needs to give more precise information about exactly what the input values represent and how these were varied. For example, does "temperature" refer to mean annual air temperature? (also see minor comments below).

Reference to unpublished data and author's PhD thesis. Where possible, either avoid reference to these materials that have not been peer-reviewed by only citing the literature, which is reviewed and accessible to the reader, or present the data referred to where this is not possible.

Minor points by line number

P1, line 16: missing 'the' - "from the central Southern Alps".

P2, 1–2: in addition to or instead of introducing the LGM in the NH, set the context in the SH.

P2, 9–13: changes in temperature are summarised by McKinnon, but changes in precipitation are not. Mention both here, making a summary of existing data for past precip change as needed.

P2, 22: give the ages assigned to the LGM here.

P4 31: whilst the authors are thanked for their citations to three of my papers, I think the citation here and elsewhere should refer to Rowan et al. 2013, Geology (glacier modelling) rather than Rowan et al. 2012, Quat Geochron (Canterbury Plains OSL).

P5, 4: misleading sentence as "other models" really only refers to different applications of the same model (Plummer and Phillips, 2003). Would be useful to mention here how these models are similar.

P5, 12 and throughout: the terms "contemporary" and "modern" are both used to refer to the observed glaciers. Both these terms are relative to the period under consideration, so slightly misleading here (e.g. could mean "contemporary to the LGM"). I suggest replacing both with "present day" throughout to avoid confusion.

P6, 4–7: the argument that geothermal heating does not affect glacier mass balance could be strengthened here, by adding a line estimating the total heat delivered to the glacier bed during a short period of volcanism and comparing this with the magnitude of other sources of energy for melting.

P6, 13–19: Although some of the present-day (recessional) glaciers are debris covered, the exclusion of rock debris cover from these simulations is a valid assumption for a study of advancing/steady state glaciers where high velocities through the entire glacier would have been likely to efficiently export debris and prevent the development of a substantial debris layer. However, the authors are investigating glaciers close to active volcanism. The impact of thin but extensive debris layers on ablation is likely to be more significant, but could be evaluated by comparison of simulated/measured glacier velocities with the intervals between eruptions to estimate the residence times of debris on the glacier ablation area surfaces. I would suggest moving this paragraph to the discussion of model uncertainties.

P6, 31: to be more precise, phrase as "mass balance imparts [the] greatest uncertainty [in simulated glacier extents]".

P7, 16: please state the lapse rates used here.

P7, 19: is the value of 2.5°C derived from the mean standard deviation in daily temperature for each month? This value seems reasonable for NZ, but it would be useful to

know how it was derived.

P7, 20: what is meant by "guide the interpolation of the station data"? How do your calculated precip grids compare to Tait's?

P7, 22: How was steady state evaluated? State the model output required for the results to be classified as steady state in the model description section.

P8, 22: Slightly confusing terminology, here and elsewhere if "step coolings" are negative values, does that not imply warming? Replace the meaning of dT with "difference in temperature from present day values" or similar, as these steps represent different temperature conditions not simulations run under a constantly changing temperature ("temperature change"). This also allows you to always refer to your simulation temperature with the same sign making the text easier to follow, so always as "difference in temperature = –2°C" rather than "dT = –2°C" and "cooling of 2°C".

P10, 13: remove "precise"

P12, 7–15: the impact of bed topography on glacier extent is interesting. Could you compare in a little more detail (i.e. quantatively) with McKinnon's results here, and suggest how other studies could carry out similar reconstructions (what is meant by "expert-defined topo reconstructions"?) and testing of bed topo?

P13, 23–25: rather awkward sentence here, rephrase.

P13, 32: you could also cite here: Schaefer et al., 2015, Quaternary Science Reviews.

P14, 32: impact of surface debris on ablation needs a citation; suggest Ostrem, 1959, Geografiska Annaler, and/or Anderson, 2000, Journal of Glaciology.

P16, 15: "a reduction in precip of [up to] 25%".

P16, Conclusions: I would prefer the conclusions to be written as continuous prose rather than numbered points. Worth mentioning here the value of these results in expanding the geographic range of glacier-based palaeoclimate reconstructions beyond

the Southern Alps.

P17, 5: avoid the use of "significantly" unless this is statistically the case.

Figure 2 caption. Under what climate condition was this simulation made?

———————————————

---

## Referee Comment (RC2) · S. Mills (Referee) · 11 Feb 2016

General comments

This paper uses a 2D glacier modelling approach to determine likely temperature and precipitation changes during LGM in central North Island, New Zealand, based on mapped moraine limits. The paper uses a step change in temperature and variations in precipitation in order to determine the most likely ice extent during the LGM, as well as studying the sensitivity of secondary parameters such as albedo. The paper also takes into account changes in the topography of the study area, as a result of post-glacial lava flows that have built up cones or infilled glacial troughs. It is concluded that temperatures during the LGM would have been in the order of 4-7 °C colder during the
[Figure]

LGM, with no change in precipitation, which is in accordance with other proxy records. This study improves understanding of past temperature change during the LGM in this relatively understudied region of North Island. The paper is well written and suggested changes are relatively few, although I do have a couple of recommendations.

Specific comments

1.) The paper would benefit from an explanation of the geomorphology of the area in more detail, including a geomorphological map outlining the key geomorphological features for each volcano. The prominent moraines are illustrated in Figure 1A, but it's very difficult to see how they relate to the topography, and their altitudinal range.

2.) The model has been run for present-day conditions with T and P =0. The authors mention that there is overestimation of ice in some of the catchments, however it would be useful if this were explored in more detail. Figure 2 seems to represent mass balance, although the figure caption suggests that this is also ice extent, however this isn't really clear. It would be useful to include a figure with modelled ice extent/thickness under present-day conditions. The authors use ice thickness from Keys (1988) to remove the ice, but how do the present-day simulations of ice thickness compare with this data? Are simulated ice thicknesses comparable to those measured? The areas where the model overestimates ice under present conditions are also where ice overspills during the temperature steps, which suggests that a more comprehensive model validation is needed. What does it take to get a better fit in the Whangaehu and Whakapapa catchments? E.g. variations in precipitation? I appreciate that the authors note the effect of Crater Lake and that modelled ice extent is better aligned in catchments that do not receive ice from this area, however this could be explored in more detail.

3.) It would be useful to look at precipitation in more detail. How many climate stations have been used in the creation of the precipitation grids and over what altitudinal range do the climate stations occur? It is later mentioned that there are no climate stations on the Tongariro massif therefore there is potentially a large uncertainty in precipitation

in these areas. I think that variations in precipitation between the two volcanoes needs to be explored in more detail or at least acknowledged much more effectively, although I note that this is mentioned briefly in page 14 lines 13-22. There also appears to potentially be asymmetry in the distribution of the moraines on Tongariro. Is this correct? It's difficult to tell without knowledge of the altitude at which these occur but it may also be something to explore in more detail.

Technical corrections

Page 4 line 21 rephrase 'moraines present outboard of these positions'.

Page 4 line 22 . . .are also supported by recent. . .

Page 7 line 1. What is the original resolution of the dem?

Page 7 line 4 How does Keys (1988) determine the ice thickness? It would be useful to give a little more information here.

Page 7 (model assessment) and reference to Figure 2. Figure caption says that the figure shows modelled steady state ice extent and mass balance but it only seems to show mass balance. It would be useful to have another figure specifically showing ice extent and thickness.

Page 10 line 35. Fig. 6a, b?

Page 14 line 6. The authors mention that there is good agreement between observed ice distribution on Mt. Ruapehu and simulations using the 30-year data set but I'm not sure that I agree with this. If figure 2 is showing the ice extent then as stated previously, it overestimates ice extent of the Whangaehu and Whakapapa glaciers and this needs to be addressed more effectively.

Figure 1. It would be really useful to know the altitudinal range of the moraines and I would suggest also including these on 1B.

Figure 4. It would be really useful to have some idea of the topography / elevation on

this figure, whether this is as a shaded dem surface or contours.

Table 1. Wahianoa should be WAH. Also, how was the ELA derived in Keys (1988)?

Table 2 Changes in lapse rates could be explored in more detail given the uncertainty in lapse rates during the LGM. What is the seasonality that has been applied? What seasonal lapse rate was originally applied and what is the justification for running the model for -0.006 °C m-1 across all months? What are contemporary lapse rates?

[Figure]

---

## Referee Comment (RC3) · Anonymous Referee #3 · 11 Feb 2016

The manuscript presents the first detailed 2-D modelling experiments from nine catchments on volcanic summits in central North Island, New Zealand. The authors conclude that regional temperatures were reduced by between 4 °C and 7 °C at the LGM when compared to present. The manuscript is well written and presented and makes an important addition to the available literature. There are some points that need additional information but overall the paper is well presented and of a high quality.

Main points:

Model simulations: 1. To ensure the LGM results are robust sensitivity tests against the modern ice distribution should be presented / discussed. Whilst the comparison to presumed LGM limits are presented, the lack of direct comparisons to modern ice

distribution does limit confidence. 2. The details and parameterization of the modelling should be made more explicit throughout

Geomorphology: 1. As it stands the geomorphological interpretation of the sites is not clear or well presented, leaving the reader somewhat unclear on the relationships. Whilst this is presented in other papers that are referenced, basic geomorphological maps and condensed chronologies should be presented to demonstrate clearly that the features being used to constrain the model simulations are clearly synchronous. As it stands there is no discussion of the possibility of multiple post glacial ice expansion, especially in the volcanic setting, where post glacial changes could make significant impacts on glacial dynamics and mass balance perhaps explaining the differences in the model simulations between the catchments? Perhaps a little more discussion and a further figure would achieve this?

Otherwise, this is a well written, and thorough piece of work that will make a valuable addition to this special edition!

---

## Author Comment (AC1) · 13 Mar 2016

We thank the three reviewers for providing thorough and constructive feedback on our manuscript. Below, we respond to the specific points made by each reviewer.

**RESPONSE TO REVIEWER ONE (RC1)**

**Main points**

[Figure]

**RC1-1:** *Model comparison with present day glaciers. The glacier model is tuned to LGM extents and the present-day simulated mass balance is shown in Fig. 2. However, a simulation of present day ice extents is not provided, which would give a much more valid comparison with the mapped ice extents presented in Fig. 2 as it is difficult for the reader to imagine the extents generated by the presented mass balance. Tuning a glacier model to a single snapshot of the glacial history is relatively easy compared to simulating multiple phases of glaciation to demonstrate that the model parameterisation is valid. The authors should present a simulation under present-day climate conditions, and if there is geomorphological evidence for glacier extents post-LGM then these simulations could also be presented even if their ages are unknown, which may be valuable to subsequent research projects.*

**Author response:** Issues concerning our comparison of modelled vs. empirical present day ice extent and mass balance, presented in Section 3.2 and Fig. 2, are echoed in various forms across the referee reports, therefore clearly must be addressed. Reviewer one suggests that a comparison of modelled vs empirical ice extent is absent from the manuscript, but should be included in order to assess model performance. However, we have already presented such a comparison in Figure 2. This figure shows the steady-state ice distribution (in two dimensions) and mass balance from a simulation forced by our modern (1981-2010 average) climate grids. In ice-covered cells, mass balance is presented coloured in red (negative) and blue (positive) diverging from white (zero), as is typically in presentation of such data. This result is plotted over a hillshade DEM (greys), which we believe provides a clear distinction between ice-covered cells and non ice-covered cells (e.g. ice extent). Also shown are the glacier outlines according to the local topographic maps (black line), which permits visual comparison of modelled and empirical ice extent.

We believe that the confusion surrounding the presence/absence of the present day comparison may arise from the limited discussion of this figure in Section 3.2. The original text in this section comprises a short statement of which modelled glaciers

agree or disagree with empirical limits (P7, lines 24-26), followed by a short discussion of the most likely causes of this disagreement (P7, lines 28-33 and P8, lines 9-20). Comments provided by all of the reviewers have made it clear this section of text lacks depth and clarity. We propose to revise this section by: (i) providing a fuller description of the model mismatches with empirical ice limits; (ii) integrating the discussion of reasons for mismatches (e.g. geothermal, wind) within this description; and (iii) providing more direct references to Fig. 2 throughout this section. As suggested by RC1 below, we will also provide a more thorough figure caption for Fig. 2, based on our description above, which more clearly describes the data presented.

**RC1-2:** *Geomorphological evidence for glacier advance. The moraine limits presented in Fig. 1 are derived from field mapping by several studies, including the thesis work of the current author. The manuscript would benefit from a more detailed, maybe separate, first figure presenting the different landforms mapped by each study and identifying landscape modification by post-LGM volcanism, to convince the reader that the geomorphological evidence used to constrain the modelling does represent (at least fairly) stable LGM glacier termini. This could also clarify what was the basis for choosing the ice margins to fit the model to, as indicated by the red lines in Figure 1. Also, a useful point to add to the aims of the paper (P2) and address; is there evidence for post-LGM moraine formation, or is it likely that this would have been removed by volcanism? Does the difference in simulated dT for each volcanic peak result from the moraines in each location representing different stages of the LGM, or did each peak show a different response to LGM cooling?*

**Author response:** In the original submission (Section 2.2 and Figure 1A) we summarise previous mapping efforts and specifically highlight the good agreement between previous investigations of late Quaternary glacial geomorphology on the volcanoes and their respective inferences of the LGM ice limits. We have recently shown such inferences to be correct in one catchment of Tongariro massif (Eaves et al., 2016.

*QSR*.). This direct dating also provides stronger grounds for morphostratigraphic correlation of undated moraines, based on characteristics such as morphology and position in the landscape, relative to the dated moraines.

Each reviewer raised concerns that our presentation of the geomorphic evidence was insufficient, which casts doubt over the veracity of our LGM palaeotemperature estimates. We propose the following revisions: (1) replace Figure 1A with a similar panel that includes the mapped moraines (differentiated between pre-, syn-, and post-LGM) and post-glacial lavas as polygons, together with the line outlines of previous reconstructions (e.g. McArthur and Shepherd, 1990; Barrell, 2011) and the LGM ice limits targeted in this study; (2) include the LGM ice limits that we targeted in panel 1B (see comments by reviewer 2); (3) provide an additional paragraph in Section 2.2 where we describe in greater detail our rationale and approach for assigning moraines to the LGM, which is based on our recently published moraine chronology, and extensive field investigations of moraine morphostratigraphy.

Concerning the latter points made by reviewer 1 above, we do explicitly consider chronological uncertainties, including the possibility of multiple LGM advances, in Section 5.1.1 'Other sources of uncertainty' (P13, lines 23-35). We do not observe evidence in the moraine record for large (c. km scale) glacier length fluctuations through the LGM. Most of the catchments we studied exhibit multiple moraine ridges about the inferred LGM positions, however the close spacing of these ridge crests (less than 1 grid cell in the model) suggests that glacier termini maintained a similar position through the LGM. We will add this statement to our revised final paragrah of Section 2.2, as described above.

**RC1-3:** *Description of the glacier model. The description of the model set up (Section 3.1.1) needs to give more precise information about exactly what the input values*

*represent and how these were varied. For example, does "temperature" refer to mean annual air temperature? (also see minor comments below).*

**Author response:**We agree that temperature must be better defined, and will seek to address the other minor points highlighted below (e.g. lapse rates, temperature variability, precipitation grids). We will also move the model input data section (previously Section 3.1.3) ahead of the mass balance model description, so that the source data and its characteristics (e.g. temporal resolution) are available alongside the description of model parameters.

**RC1-4:** *Reference to unpublished data and author's PhD thesis. Where possible, either avoid reference to these materials that have not been peer-reviewed by only citing the literature, which is reviewed and accessible to the reader, or present the data referred to where this is not possible.*

**Author response:**The unpublished Ar/Ar lava age dataset of co-author Conway has now been submitted for publication, therefore we will include this submitted manuscript in the reference list so that readers have the opportunity to find the data in the near future. In two instances we have cited the thesis of first author Eaves. We do not rely on the data within this document to draw our conclusions, but merely seek to highlight that pertinent information exists in this document. This thesis has undergone examination by three reviewers and is publically available, therefore we feel justified in including this reference.

**Minor points by line number**

**P1, line 16:** *missing 'the' - "from the central Southern Alps"*
**Author response:**We agree.

**P2, 1–2:** *in addition to or instead of introducing the LGM in the NH, set the context in the SH.*
**Author response:**We agree there is a disconnect between the opening two sentences and the remainder of the opening paragraph. We suggest that a short opening paragraph that first defines the LGM, and outlines the importance of this interval as a target for palaeoclimate studies. Our following paragraph can then focus in on the LGM in the Southern Hemisphere and New Zealand.

**P2, 9–13:** *changes in temperature are summarised by McKinnon, but changes in precipitation are not. Mention both here, making a summary of existing data for past precip change as needed*
**Author response:**We agree that mention of precipitation is lacking. We propose a short sentence highlighting that palaeoprecipitation estimates are even more scarce, and often qualitative.

**P2, 22:** *give the ages assigned to the LGM here.*
**Author response:**We agree.

**P4 31:** *whilst the authors are thanked for their citations to three of my papers, I think the citation here and elsewhere should refer to Rowan et al. 2013, Geology (glacier modelling) rather than Rowan et al. 2012, Quat Geochron (Canterbury Plains OSL).*
**Author response:**We agree.

**P5, 4:** *misleading sentence as "other models" really only refers to different applications of the same model (Plummer and Phillips, 2003). Would be useful to mention here how these models are similar.*
**Author response:**We agree and will expand this sentence accordingly.

**P5, 12 and throughout:** *the terms "contemporary" and "modern" are both used to refer to the observed glaciers. Both these terms are relative to the period under consideration, so slightly misleading here (e.g. could mean "contemporary to the LGM"). I suggest replacing both with "present day" throughout to avoid confusion.*
**Author response:**We agree and will remove the use of 'contemporary' to avoid confusion.

**P6, 4–7:** *the argument that geothermal heating does not affect glacier mass balance could be strengthened here, by adding a line estimating the total heat delivered to the glacier bed during a short period of volcanism and comparing this with the magnitude of other sources of energy for melting*
**Author response:**We prefer to cite empirical studies of these effects to support this point.

**P6, 13–19:** *Although some of the present-day (recessional) glaciers are debris covered, the exclusion of rock debris cover from these simulations is a valid assumption for a study of advancing/steady state glaciers where high velocities through the entire*

*glacier would have been likely to efficiently export debris and prevent the development of a substantial debris layer. However, the authors are investigating glaciers close to active volcanism. The impact of thin but extensive debris layers on ablation is likely to be more significant, but could be evaluated by comparison of simulated/measured glacier velocities with the intervals between eruptions to estimate the residence times of debris on the glacier ablation area surfaces. I would suggest moving this paragraph to the discussion of model uncertainties.*

**Author response:**This is a methodological decision, thus we believe that it requires justification in this section of the manuscript. We note that the possible implications of this decision are considered, in light of the results, in Section 5.1.1. (P14, lines 30-35).

**P6 31:** *to be more precise, phrase as "mass balance imparts [the] greatest uncertainty [in simulated glacier extents]"*
**Author response:**We agree.

**P7, 16:** *please state the lapse rates used here*
**Author response:**We agree.

**P7, 19:** *is the value of 2.5° C derived from the mean standard deviation in daily temperature for each month? This value seems reasonable for NZ, but it would be useful to know how it was derived.*
**Author response:**Yes, this is correct. We derived this value from analysis of the 30yr (1981-2010) daily mean temperature data from the climate stations at Whakapapa Village (1097 m asl), which is situated close to the centre of the model domain. We will include this information in the revision.

**P7, 20:** *what is meant by "guide the interpolation of the station data"? How do your calculated precip grids compare to Tait's?*

**Author response:** Our description here explicitly cites the precipitation interpolation method described by Anderson and Mackintosh (2012), which states:

*'For each day, the proportion of mean annual precipitation at each station was interpolated across the model grid, and this proportional value was then multiplied by the mean annual precipitation surface to calculate the precipitation at each grid cell.'*

The only difference being that our calculations for the central North Island were performed at monthly resolution, as opposed to daily. While being mindful of the journal guidelines to avoid replicating descriptions of published methods, we realise this could be explanded upon. We will provide a concise description of this method in our revision. See also point RC2-3, below.

**P7, 22:** *How was steady state evaluated? State the model output required for the results to be classified as steady state in the model description section.*

**Author response:** This information is currently included in Section 3.3.1 ('Equilibrium is achieved when the rate of ice volume change becomes close to zero, which takes 200-350 model years depending on the magnitude of $\Delta T$.'). We will include a global statement below the 'Experimental design' heading, which outlines this fact, and reports the model time steps.

**P8, 22:** *Slightly confusing terminology, here and elsewhere if "step coolings" are*

*negative values, does that not imply warming? Replace the meaning of dT with "difference in temperature from present day values" or similar, as these steps represent different temperature conditions not simulations run under a constantly changing temperature ("temperature change"). This also allows you to always refer to your simulation temperature with the same sign making the text easier to follow, so always as "difference in temperature = –2° C" rather than "dT = –2° C" and "cooling of 2° C".*
**Author response:**We agree. We will alter the terminology to 'temperature change' in order to avoid confusion.

**P10, 13:** *remove "precise"*
**Author response:**We agree.

**P12, 7–15:** *the impact of bed topography on glacier extent is interesting. Could you compare in a little more detail (i.e. quantatively) with McKinnon's results here, and suggest how other studies could carry out similar reconstructions (what is meant by "expert-defined topo reconstructions"?) and testing of bed topo?*
**Author response:**Our 'expert-defined topographic reconstructions' are described in Section 3.3.3. 'Experiment 3: Palaeo-topography' (P9 lines 7-26). These reconstructions comprise digital elevation models of both volcanoes, which have been created by manually altering the present day contour lines to remove or replace topography that has been added or removed by post-glacial volcanism (e.g. lava flows, explosion craters). These alterations are informed by the results of a recent mapping and dating campaign, as well as prior investigations - see references cited in manuscript. We realise that we only use the term 'expert-defined topographic reconstructions' in the discussion, therefore the connection to Section 3.3.3 is not obvious. We will change this phrasing to be consistent with the methodology.

Most of the topographic changes we prescribed resulted in a reduction of present day surface elevations (i.e. removal of lava flows), and were on the order of 50-300 m. Surface air temperatures in our climate grids are therefore c. 0.3 - 1.5 °C warmer in regions with lower topography, thus a greater temperature forcing is generally required to simulate the LGM glaciers, when compared to simulations that use present day topography. In one catchment our prescribed bed changes resulted in increased ice flux from the central accumulation zone, which lowered the temperature forcing required to simulate the LGM limits.

In modelling the LGM Pukaki Glacier, McKinnon et al. (2012) were faced with a similar problem regarding post-glacial topographic change. In this instance, post-glacial sedimentation within Lake Pukaki masks the bed topography of the former glacier. McKinnon et al. used constraints from seismic imaging and moraine-based reconstructions of the LGM glacier surface to estimate the bed topography for early and late LGM times, using glacier flow modelling. This work also found that lowering of the glacier bed occurred through the LGM, caused by glacial erosion, which may have contributed to the glacier length change indicated by the moraine record.

We are reluctant to draw quantitative comparisons between our study and that of McKinnon et al. because it is difficult to deconvolve the impacts of the different approaches (e.g. methods of bed reconstruction, types of flow models) and site-specific characteristics (e.g. hypsometry). Rather, we prefer to reinforce the point that palaeo-topographies may impart small, but non-trivial impacts on palaeoclimate reconstructions, and that this uncertainty can be investigated where geological evidence is available to inform the sign and magnitude of bed elevation changes. We will highlight this point more clearly in our revised manuscript.

[Figure]

**P13, 23–25:** *rather awkward sentence here, rephrase*
**Author response:** We agree. Alternative: 'Only two of the catchments studied here (MPO, WAH) have moraines been directly dated to the LGM...'

**P13, 32:** *you could also cite here: Schaefer et al., 2015, Quaternary Science Reviews.*
**Author response:** We agree.

**P14, 32:** *impact of surface debris on ablation needs a citation; suggest Ostrem, 1959, Geografiska Annaler, and/or Anderson, 2000, Journal of Glaciology.*
**Author response:** We agree. Ostrem (1959) is a classic, empirical example of the point made here.

**P16, 15:** *"a reduction in precip of [up to] 25%".*
**Author response:** We agree.

**P16, Conclusions:** *Conclusions: I would prefer the conclusions to be written as continuous prose rather than numbered points. Worth mentioning here the value of these results in expanding the geographic range of glacier-based palaeoclimate reconstructions beyond the Southern Alps.*
**Author response:** We agree, the numbers are almost redundant given the length of the text associated with each. We will restructure these paragraphs and include the reviewers suggestion here.

**P17, 5:** *avoid the use of "significantly" unless this is statistically the case.*
**Author response:**We agree.

**Figure 2 caption:** *Under what climate condition was this simulation made?*
**Author response:**This is the output from the present day (no change in temperature or precipitation). As stated in RC1-1, we will revise this figure caption to avoid confusion.

**RESPONSE TO REVIEWER TWO (RC2)**

**Specific comments**

**RC2-1:** *The paper would benefit from an explanation of the geomorphology of the area in more detail, including a geomorphological map outlining the key geomorphological features for each volcano. The prominent moraines are illustrated in Figure 1A, but it's very difficult to see how they relate to the topography, and their altitudinal range.*
**Author response:** We refer to our response to RC1-2, above.

**RC2-2:** *The model has been run for present-day conditions with T and P =0. The authors mention that there is overestimation of ice in some of the catchments, however it would be useful if this were explored in more detail. Figure 2 seems to represent mass balance, although the figure caption suggests that this is also ice extent, however this isn't really clear. It would be useful to include a figure with modelled ice extent/thickness under present-day conditions. The authors use ice thickness from Keys (1988) to remove the ice, but how do the present-day simulations of ice thickness compare with this data? Are simulated ice thicknesses comparable to those measured? The areas where the model overestimates ice under present conditions are also where ice overspills during the temperature steps, which suggests that a more comprehensive model validation is needed. What does it take to get a better fit in the Whangaehu and Whakapapa catchments? E.g. variations in precipitation? I appreciate that the authors note the effect of Crater Lake and that modelled ice extent is better aligned in catchments that do not receive ice from this area, however this could be explored in more detail.*
**Author response:** We refer the reader to our response to reviewer one (RC1-1) concerning the comparison of modelled and empirical two dimensional ice extent. In addition, reviewer 2 considers ice thickness as a possible metric for model performance. The ice thickness data of Keys (1988), which we use to create an 'ice-free'

topography, is derived from a combination of crevasse depth measurements on the glaciers, and an ice penetrating radar survey on the summit plateau region. For most of the small cirque glaciers the thickness is based on the crevasse measurements. Thus, these data provide only minimum-limiting estimates. Given this relatively poor empirical constraint of present day ice thickness, we do not believe a comparison of this parameter would provide robust insight to model performance. However, we do recognise that underestimation of ice thickness may cause us to overestimate bed elevation, thus giving temperatures on the upper mountain that are too low and, consequently, mass balance values that are too high. This is a further possible contributor to overestimation of ice extent in some catchments and should be recognised in our revision of Section 3.2 (see RC1-1 for more details on this proposed revision).

**RC2-3:** *It would be useful to look at precipitation in more detail. How many climate stations have been used in the creation of the precipitation grids and over what altitudinal range do the climate stations occur? It is later mentioned that there are no climate stations on the Tongariro massif therefore there is potentially a large uncertainty in precipitation in these areas. I think that variations in precipitation between the two volcanoes needs to be explored in more detail or at least acknowledged much more effectively, although I note that this is mentioned briefly in page 14 lines 13-22. There also appears to potentially be asymmetry in the distribution of the moraines on Tongariro. Is this correct? It's difficult to tell without knowledge of the altitude at which these occur but it may also be something to explore in more detail.*

**Author response:** We use daily precipitation data interpolated across the model domain from c. 40 climate stations distributed across New Zealand to provide the proportion of annual precipitation (as given by the mean annual precipitation surface) that falls each month. Thus, the station data provides the monthly distibution of precipitation, while the absolute values are dictated by the mean annual surface. The seasonal distribution of precipitation in grid cells situated far from any climate stations

is therefore less well constrained by empirical data than that for cells situated closer to stations. We used data from nineteen climate stations situated within 50 km of the centre of the model domain, which range in altitude from 125 m to 1097 m above sea level. Most of these stations only have partial coverage between 1981-2010, but the spatiotemporal distribution provided by all stations used is sufficient to provide an interpolated surface for the model domain for all years. We note that climate data from Whakapapa village, which is the highest elevation station (1097 m) and is situated close to the centre of the model domain, has continuous coverage during the 1981-2010 interval.

The absence of precipitation measurements from the former accumulation zones of the LGM glaciers (i.e. $> 1500$ m asl) on either volcano means that (i) the sub-annual precipitation distribution is less well constrained for these regions, and (ii) we lack observational data with which to evaluate the absolute precipitation totals provided by the mean annual precipitation surface. These limitations are unfortunate, but are present in all such studies. On page 14 lines 1-22, we consider the possible implications of inaccurate present/past precipitation distirbutions for our palaeotemperature results.

There is no obvious east-west asymmetry in moraine distribution on either volcano. Former ice limits on the north - north-western slopes of Tongariro massif are not well preserved due to post-glacial volcanism (i.e. North Crater - box 7 in Figure 3), which may hinder identification of any climatic-based asymmetry in palaeo-ice distribution. Our proposed revisions to Figure 1, as stated in RC1-2 above, we help to illustrate this point.

**Technical corrections**

**Page 4 line 21** *rephrase 'moraines present outboard of these positions'.*
**Author response:**We will change to '...correlation of outboard moraines to the LGM.'

**Page 4 line 22** *...are also supported by recent...*
**Author response:**We agree.

**Page 7 line 1** *What is the original resolution of the dem?*
**Author response:**15 m. We will add this information.

**Page 7 line 4** *How does Keys (1988) determine the ice thickness? It would be useful to give a little more information here.*
**Author response:**We refer the reader to our response to RC2-2 above.

**Page 7 (model assessment) and reference to Figure 2** *Figure caption says that the figure shows modelled steady state ice extent and mass balance but it only seems to show mass balance. It would be useful to have another figure specifically showing ice extent and thickness.*
**Author response:**We refer the reader to our responses to RC1-1 and RC2-2 above.

**Page 10 line 35** *Fig. 6a, b?*
**Author response:**Yes, we will revise.
**Page 14 line 6** *The authors mention that there is good agreement between observed ice distribution on Mt. Ruapehu and simulations using the 30-year data set but I'm not sure that I agree with this. If figure 2 is showing the ice extent then as stated previously, it overestimates ice extent of the Whangaehu and Whakapapa glaciers and this needs to be addressed more effectively*
**Author response:**We agree that the fit in Whangaeuhu and Whakapapa is poor, relative to the other catchments. We note this poor fit on P7 line 25-26, then go on to discuss that this is likely because we neglect the energy provided by the geothermally-heated Crater Lake (P7 lines 28-34). This lake is thought to have been initiated during the Holocene, therefore we do not try to include this in our LGM model experiments (P8 lines 4-7). Failure to capture complex snow accumulation patterns caused by wind redistribution may also be an important factor causing the poor fit in these catchments (P8 lines 9-20).

As stated in the response to RC1-1, we believe this section could be restructured to clarify the above points. We will take the reviewers concern into consideration when revising this text.

**Figure 1** *It would be really useful to know the altitudinal range of the moraines and I would suggest also including these on 1B*
**Author response:**We refer to our response to RC1-2, above.

**Figure 4** *It would be really useful to have some idea of the topography / elevation on this figure, whether this is as a shaded dem surface or contours.*
**Author response:**The purpose of the figure is to show the simulated ice extent against

the mapped LGM ice limits, in order for assess the likeness of the simulated ice mass to the geologically-inferred limits. We have experimented with different presentation of this data and found that the plain background best serves our purpose. Inclusion of other datasets detract from this comparison. Thus, we prefer for this figure to remain as is.

**Table 1** *Wahianoa should be WAH. Also, how was the ELA derived in Keys (1988)?*
**Author response:**We will amend the glacier code. Glacier ELAs, as reported by Keys (1988) were determined by repeat ground/aerial surveys of the end of summer snowline. We will note this method in the second paragraph of Section 2.1, where this survey is reported.

**Table 2** *Changes in lapse rates could be explored in more detail given the uncertainty in lapse rates during the LGM. What is the seasonality that has been applied? What seasonal lapse rate was originally applied and what is the justification for running the model for -0.006 °C m-1 across all months? What are contemporary lapse rates?*
**Author response:**We agree that there is high uncertainty in prescribing temperature lapse rates for the past. The spatial and temporal pattern of present day lapse rates on the central North Island volcanoes are poorly constrained, being limited by the paucity of climate stations. Temperature lapse rates for the LGM are unknown. In absence of well-distributed present day temperature measurements on the volcanoes, we use the seasonal, upland lapse rates of Norton (1985) - see Table 1 below. These values are derived from multiple linear regression of multi-decadal temperature measurements from 71 climate stations in New Zealand situated >300 m elevation. The seasonality exhibited in this dataset is thought to reflect the higher propensity for inversions during autumn/winter. We note that the upland values are within c. 0.3°C of

**Table 1.** Mean seasonal upland ($>$ 300 m) air temperature lapse rates of Norton (1985)

| Month | Lapse rate |
|---|---|
| DFJ | -5.3 °C km$^{-1}$ |
| MAM | -4.9 °C km$^{-1}$ |
| JJA | -4.8 °C km$^{-1}$ |
| SON | -5.7 °C km$^{-1}$ |
| Mean annual | 5.1°C km$^{-1}$ |

those derived from a much larger dataset (*n*=301) that also includes lowland stations. We consider this dataset the best available for constraining present day vertical temperature variation in New Zealand, and it is commonly employed for interpolation of present day temperature data (e.g. Tait and Macara, 2014, *Weather and Climate*) and palaeoclimate applications (e.g. Golledge et al., 2012, *QSR*; Putnam et al., 2013, *QSR*).

We ran alternative experiments with a uniform lapse rate of -6°C km$^{-1}$ mainly for comparative reasons, as several other LGM glacier modelling studies employ this value (e.g. Rowan et al., 2012, *Geol.*; Putnam et al., 2013, *EPSL.*). We found that the -6°C km$^{-1}$ experiments altered the palaeotemperature results by $<$0.2°C. This result mirrors the findings of Putnam et al. (2013) *QSR*, who conducted a similar experiment. In general, the effects of altering the temperature lapse rate are predictable. Greater values result in smaller palaeotemperature anomalies for palaeoglacier simulations, because the lower surface temperatures mean less energy is available for melt and there is increased potential for solid precipitation. And vice versa for lower lapse rate values. This effect is illustrated by our sensitivity experiment (Figure 5). Altering the seasonality of lapse rates for the LGM would make for more interesting experiments, but these experiments should have some physical basis (e.g. global climate model simulations that use LGM boundary conditions). Given the current uncertainties associated with both

present and past lapse rates, such experiments are beyond the scope of this present study.

[Figure]

**RESPONSE TO REVIEWER THREE (RC3)**

**Main points**

**RC3-1:** *Model simulations. To ensure the LGM results are robust sensitivity tests against the modern ice distribution should be presented / discussed. Whilst the comparison to presumed LGM limits are presented, the lack of direct comparisons to modern ice distribution does limit confidence. 2. The details and parameterization of the modelling should be made more explicit throughout.*

**Author response:** We believe our responses to comments RC1-1 and RC2-2 (above) sufficiently address the first point made here.

**RC3-1:** *Geomorphology. As it stands the geomorphological interpretation of the sites is not clear or well presented, leaving the reader somewhat unclear on the relationships. Whilst this is presented in other papers that are referenced, basic geomorphological maps and condensed chronologies should be presented to demonstrate clearly that the features being used to constrain the model simulations are clearly synchronous. As it stands there is no discussion of the possibility of multiple post glacial ice expansion, especially in the volcanic setting, where post glacial changes could make significant impacts on glacial dynamics and mass balance perhaps explaining the differences in the model simulations between the catchments? Perhaps a little more discussion and a further figure would achieve this?*

**Author response:** We refer to our response to RC1-2, above.

---

## Author Response (AR1)

cp-2016-1: Author revisions

Following the editorial comments (dated 16 Mar 2016), we have revised our manuscript in accordance with our 'Author response to reviewer comments' (AC1; 13 Mar 2016).

We have made one alteration to one of the revisions proposed in AC1. The original Figure 1 has now been split into two separate figures (now Figure 1 and Figure 2). This change is largely to improve the clarity of the revised Figure 1, which now includes geological information as requested by reviewers. The revised Figure 2 has the topographic data from panel B of the original Figure 1, as well as extra panels showing climatic data over the model domain.

We have also made minor typographical changes, not included in AC1, in order to bring the manuscript into accordance with the journal conventions. A fully marked-up manuscript is included below, which documents all changes made from the original submission.

[revised manuscript text omitted]